# Microwave Limb Sounder (MLS) observations of biomass-burning products in the stratosphere from Canadian forest fires in August 2017

Hugh C. Pumphrey[1], Michael J. Schwartz[2], Michelle L. Santee[2], George P. Kablick III[3], Michael D. Fromm[3], and Nathaniel J. Livesey[2]

[1]School of GeoSciences, The University of Edinburgh, Edinburgh, UK
[2]NASA Jet Propulsion Laboratory, California Institute of Technology, Pasadena, CA, USA
[3]US Naval Research Lab, Washington DC, USA

**Correspondence:** Hugh Pumphrey (Hugh.Pumphrey@ed.ac.uk)

**Abstract.**

Forest fires in British Columbia in August 2017 caused a pyrocumulonimbus event that injected a polluted airmass into the lower stratosphere. The Microwave Limb Sounder (MLS) on the Aura satellite first observed the polluted airmass on 14 August 2017 and continued to observe it for 60 days (100 days in water vapour). We estimate the mass of CO injected into the stratosphere to be 2400 Gg. Events in which a fire injects its burning products directly into the stratosphere are rare: this is the third of four such events in the 16 years since the launch of Aura, the second-largest of the four events, and the only one in the Northern Hemisphere. The other three events occurred in Australia in December 2006, February 2009 and from December 2019 to January 2020. Unlike the 2006 and 2009 events, but like the 2019–20 event, the polluted airmass described here had a clearly elevated water vapour content: between 2.5 and 5 times greater than that in the surrounding atmosphere. We describe the evolution of the polluted airmass, showing that it rose to an altitude of about 24 km (31 hPa) and divided into several identifiable parts. In addition to CO and $H_2O$, we observe enhanced amounts of HCN, $CH_3CN$, $CH_3Cl$ and $CH_3OH$ with mixing ratios in the range to be expected from a variety of measurements in other biomass-burning plumes. We use back-trajectories and plume-dispersion modelling to demonstrate that the pollutants observed by MLS originated in the British Columbia fires, the likeliest source being at 53.2°N, 121.8°W at 05:20 UTC on 13 August 2017.

## 1 Introduction

Fires are an important natural process in many ecosystems, forests in particular (Bowman et al., 2009). The intensity and frequency of fires can be sensitive to human intervention (e.g. forestry practices), even when humans have not provided the source of ignition (Williams and Abatzoglou, 2016). The occurrence of fires is also sensitive to climate change; we can expect

fires to become more frequent and/or more damaging in many parts of the world as the climate continues to warm (Williams et al., 2019).

The Microwave Limb Sounder (MLS) instrument (Waters et al., 2006) observes several chemical species emitted by forest fires. In tropical regions these tend to be transported to altitudes observable by this instrument by the wide-scale circulation of the atmosphere, and particularly by the strong convection along the intertropical convergence zone (Pumphrey et al., 2018).

Outside of the tropics it is rare for MLS to observe the immediate effect of fires; we describe one such event in this paper.

The summer of 2017 saw the most destructive forest fires on record in British Columbia, Canada (Government of British Columbia, 2020). On 12 August 2017 a particularly intense cluster of fires gave rise to a strong convective event that lofted biomass-burning product gases to an altitude of 10–11 km (215 hPa), as described in detail by Peterson et al. (2018), who refer to them as the "Pacific Northwest Event". We adopt this name and abbreviate it to PNE. The polluted airmass was transported

around the world, ascending to an altitude of 19 km (68 hPa) over a period of 18 days (Khaykin et al., 2018). Lidar observations of extremely high levels of aerosol in the polluted airmass have been reported from Europe (Ansmann et al., 2018; Khaykin et al., 2018; Hu et al., 2019; Baars et al., 2019) and Russia (Zuev et al., 2019). Khaykin et al. (2018) describe the PNE in detail and also report space-based lidar data from the Cloud-Aerosol Lidar with Orthogonal Polarization (CALIOP) instrument (Winker et al., 2009), showing the global evolution of the plume. Further space-based observations of aerosol from the event

are described by Kloss et al. (2019) and Torres et al. (2020). Lestrelin et al. (2021) combine satellite aerosol measurements and ERA5 (Hersbach et al., 2020) reanalysis data to provide the most detailed description to date of the evolution in time and space of the polluted airmass. Fromm et al. (2021) consider the first few days of the event in more detail, providing an estimate of the aerosol mass. Transport and radiative impacts of the event are described by Das et al. (2021).

MLS on Aura, and its predecessor on the UARS mission, have previously observed biomass burning products from intense

fire events (Livesey et al., 2004; Pumphrey et al., 2011). We report here observations made by MLS of some of the gas-phase components of the polluted airmass from the PNE. The ozone and water vapour data have been briefly mentioned by Yu et al. (2019) in a paper describing observations of aerosols from the PNE and the modelling of photochemical processes within the polluted airmass. Here we report in more detail on the full suite of biomass-burning products observed by MLS: CO, $CH_3CN$, HCN, $CH_3Cl$ and $CH_3OH$. We also describe any changes observed in the polluted airmass for all other species observed by

the instrument.

## 2   Data from MLS

The MLS instrument (Waters et al., 2006) was launched in July 2004 on NASA's Aura satellite and has operated almost continuously since then. Aura is in a polar orbit with a 98° inclination angle. The MLS instrument views in the direction of travel; as a result the measurement locations lie close to the orbit track and cover the latitude range 82° S to 82° N. Once

processed, the data consist of 3495 vertical profiles per day, spaced at intervals of 167 km along the orbit track. Estimated quantities include temperature, geopotential height, and the mixing ratios of 16 chemical species, including $H_2O$ (Lambert et al., 2007), CO (Pumphrey et al., 2007; Livesey et al., 2008) and HCN (Pumphrey et al., 2018). Data are reported on pressure

levels; for most products these are spaced at 6 levels per pressure decade, giving a representation with a vertical resolution of about 2.7 km. For most products the true vertical resolution is similar to this spacing, but for species with low mixing ratios and/or weak spectral lines it may be considerably poorer than this. For a few species ($H_2O$, $O_3$) the data are reported at a resolution of 12 levels per pressure decade. We use version 4 of the MLS data; full details on use of the data may be found in Livesey et al. (2020).

## 3 Results

### 3.1 Observations

#### 3.1.1 Carbon Monoxide

MLS CO data are recommended (Livesey et al., 2020) for use at pressure levels from 215 hPa ($\sim 11$ km) up to 0.0046 hPa ($\sim$80 km). Data are reported at the 316 hPa level ($\sim$8 km), but the CO averaging kernels in this region of the profile are not sharply peaked, so the 316 hPa measurement contains substantial contributions from the 215 and 100 hPa retrieval levels. Thus the 316 hPa data are not recommended for quantitative work (Livesey et al., 2020). The 316 hPa CO data may, to some extent, be a useful qualitative indicator of excess CO somewhere in the 8–10 km altitude range; we show the 316 hPa data in some of the figures in this paper, but do not use them to derive any quantitative results.

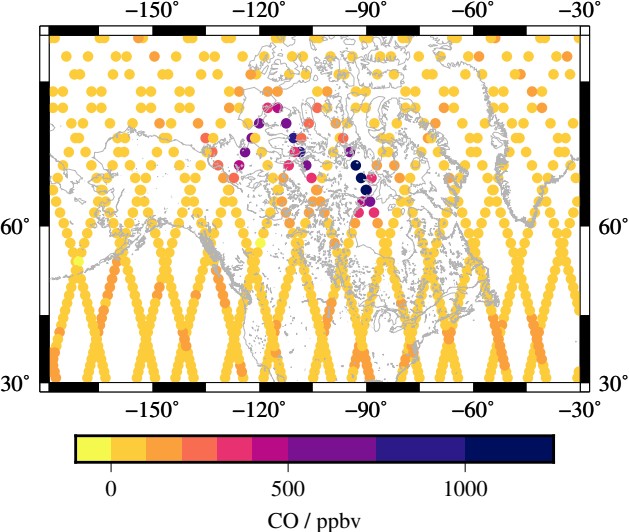

**Figure 1.** MLS measurements of CO (in ppbv) at 215 hPa ($\approx 11$ km altitude) on 14–15 August 2017. These are the first two days after the PNE on which MLS observed the enhanced CO values.

Obviously enhanced values of CO from the Pacific Northwest Event were first observed on 14 August 2017. Fromm et al. (2021) report enhanced values on 13 August: this enhancement was discovered using co-located CALIOP data and is at the

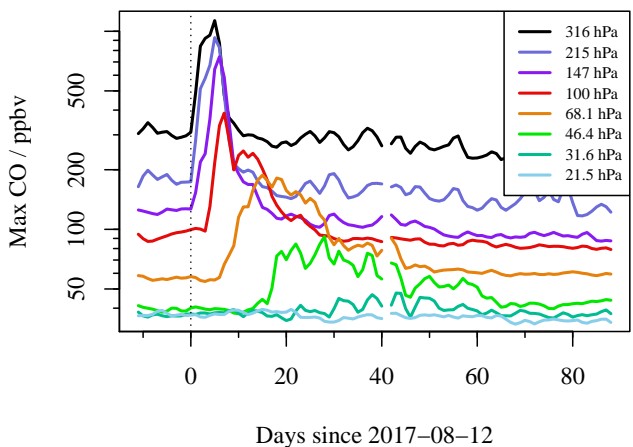

**Figure 2.** Time series of maximum CO values observed polewards of $27.5°$ N. The values shown are averages of the ten highest observed values on each day between $27.5°$ N and $83°$ N. A 3-day smoothing has been applied to each curve.

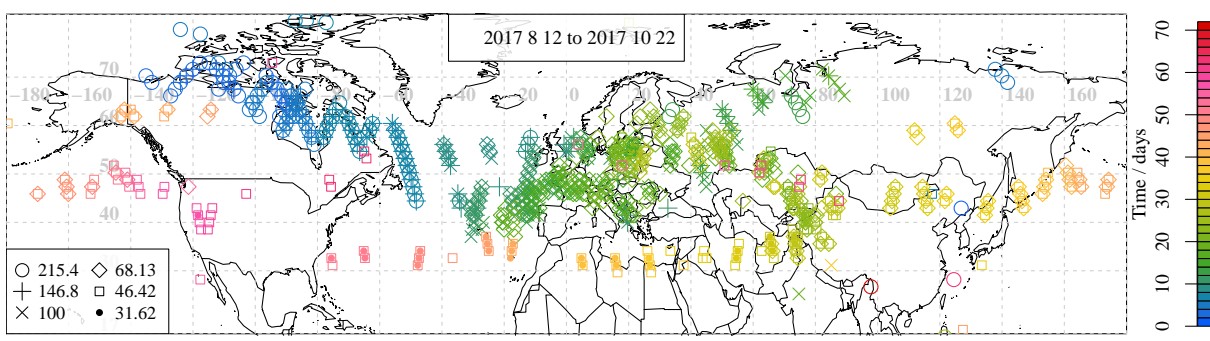

**Figure 3.** Map showing the locations of MLS observations of enhanced CO (defined as described in the text). Colours show time in days after 12 August 2017. Symbols show pressure level in hPa. The MLS measurements lie along the satellite orbit tracks; the tracks are particularly noticeable over North America and can also be seen in Fig. 1.

upper end of the normal range of values. Figure 1 shows MLS CO over North America at this time, at 215 hPa. The maximum values seen here are about 1.5 ppmv; this is about 24 times the mean value of 62 ppbv and 6–8 times the typical daily maximum values for the $27.5°$N–$82°$N latitude band for 8–11 August 2017. Figure 2 shows a time series of the maximum CO values observed at several pressure levels. It is clear that the polluted airmass reached 215 hPa almost immediately. From there, it ascended fairly rapidly to 100 hPa and then more slowly from 100 hPa to 46 hPa.

To summarise the whole event, we applied the strategy used by Pumphrey et al. (2011) to detect values that exceed the usual spread expected for a given latitude band. We found the mean value and standard deviation for all measurements in a given latitude band and at a given pressure level. We then describe the CO at a point as "enhanced" if its mixing ratio was more than a chosen number, $\alpha$, of standard deviations above the mean. The value chosen for $\alpha$ was different for each level: $\alpha = 5.7$ at 215 hPa, $\alpha = 4.6$ at 147 and 100 hPa and $\alpha = 3.9$ at all other pressure levels. These choices are made in order to show the

interesting events while rejecting almost all of the background points. An iterative approach is used to ensure that the mean and standard deviation are calculated from the unaffected points; there are few enough affected points that this calculation converges very quickly.

Figure 3 shows the locations where enhanced CO was detected. The plume is first observed over northwestern Canada, travelling southeastwards and then eastwards across northern Canada and then across the Atlantic to Europe. The behaviour of the plume while over Eurasia is more complex. After 30 days, one part of it reaches northwestern China, where it appears to divide into two parts, one heading east across the Pacific at about 47° N and the other returning westwards across the Atlantic at about 33° N.

As the polluted airmass moves far more rapidly in the zonal than the meridional direction, it is helpful to plot it against time and longitude, as in Fig. 4. The ascent of the polluted airmass (shown by the changing colour and shape of the points) is

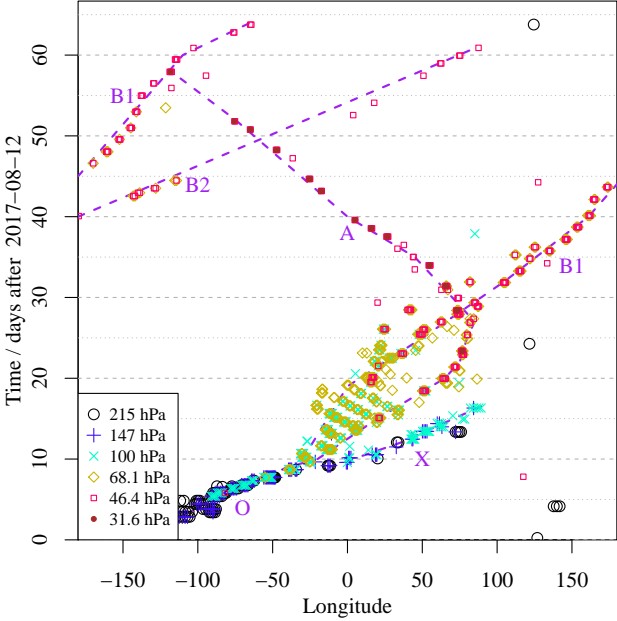

**Figure 4.** Longitudes of enhanced CO (as defined in the text), plotted as a function of time. Colours and symbols represent pressure. Dashed lines and labels identify the various parts into which the polluted airmass is observed to divide.

clearly visible in this figure. It can also be seen in Fig. 4 that the polluted airmass is divided into two parts on three occasions: at approximately 9 days, 16 days and 41 days after 12 August 2017. We label the parts in Fig. 4 using the nomenclature introduced by Lestrelin et al. (2021), with the exception of a part which Lestrelin et al. (2021) do not identify and which we label "X". After the first division, part X remains at 147 to 100 hPa, travels more rapidly than the other parts and is last seen on day 16 near 80° E, 70° N. The rest of the polluted airmass is at 100 to 68 hPa at day 9 and continues to ascend. It appears to divide into two parts (A and B1) somewhere around day 16; these two parts appear in Fig. 4 to cross over at around day 30, after which part B1 continues eastward and part A travels westward. The two parts are at different latitudes and altitudes at this time. Part

B1 lies poleward of 42°N, between 68 and 46 hPa; it travels eastward from northern China across the Pacific Ocean. Part A is equatorward of 38°N, between 46 and 31 hPa (21 to 24 km); it moves westward across northern Africa and the Atlantic Ocean.

Around day 41, part B1 appears to divide into two further parts while at a longitude near 180°E. The MLS data are too sparse to determine the location where the split occurred with any accuracy. The faster-travelling part (B2) is last seen on day 60 at 90° E; the remainder (which we continue to label B1) is last seen on day 63 at 60° W. Although the MLS CO data do not make this clear, Lestrelin et al. (2021) show that B2 actually splits apart from B1 within a few days of day 25, with B1 and B2 travelling at similar longitudes until a time near day 41.

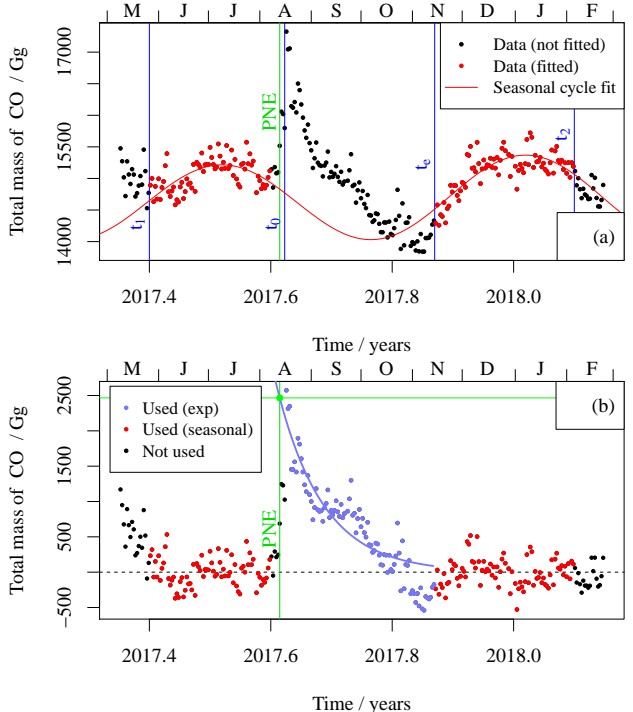

**Figure 5.** (a) Total mass of CO observed by MLS poleward of 27.5° N and at altitudes between 215 hPa and 31 hPa. The four times $t_1, t_0, t_e$ and $t_2$ are discussed in the text. The seasonal fit includes both annual and semiannual components; the annual component is much smaller in this case. (b) As (a), but with the seasonal fit subtracted from the data, and an exponential fitted to the measurements made after the fire. Both the exponential fit and the data used to construct it are shown in blue.

We can obtain an estimate of the total injected mass of CO by integrating the MLS data into partial column (215 hPa to 31 hPa) values and summing over all latitudes affected by the event, weighting each latitude band by its area. The seasonal behaviour of CO is fitted with a mean value and annual and semiannual sinusoids as shown in Fig. 5(a); this fit ignores the data for a period of 93 days after the PNE. Subtracting this first fit from the data leaves a fairly featureless time series apart from a

**Table 1.** Summary of the injected masses of CO due to the four fire events noted in the text, showing the best estimate of the total mass, with the pressure level in the "Max $p$" column being the highest pressure (lowest altitude) level included. Contributions to the error in the mass estimate caused by the choice of the parameters $t_0$, $t_e$, $t_1$ and $t_2$ are shown. Also shown is a typical error as returned by the R function `nls()`. The total error is obtained by summing the five individual errors in quadrature. This procedure is not ideal as most of the errors will be correlated with each other. However, it gives a reasonable conservative estimate. Units are Gg for all data.

| Event | Max $p$ / hPa | Mass | Error `nls()` | Error $t_0$ | Error $t_e$ | Error $t_1$ | Error $t_2$ | Total Error |
|-------|-------|------|--------|------|------|------|------|-------|
| ANY | 215 | 7700 | 220 | 300 | 40 | 30 | 40 | 400 |
| PNE | 215 | 2450 | 150 | 90 | 100 | 56 | 6 | 210 |
| BS | 147 | 1350 | 200 | 200 | 40 | 20 | 80 | 300 |
| GD | 215 | 1520 | 150 | 230 | 30 | 13 | 25 | 280 |

sudden increase following the fire and a decay after that; we fit the decay with an exponential:

$$M = M_0 \exp\left(-\frac{t}{t_d}\right)$$

The exponential curve is shown in Fig. 5(b). The estimate of the injected mass, $M_0$, is $2450 \pm 150$ Gg, with an estimated decay time $t_d$ of $28 \pm 2$ days; the errors here are those given by the `nls()` function in R (R Core Team, 2018). The `nls()` function determines the least-squares estimates of the parameters of a nonlinear model, using the Gauss-Newton method. The errors are calculated as they are for a linear least-squares model, which is a useful approximation unless the errors are very large or the problem is very nonlinear in the vicinity of the least-squares estimate. We consider this problem to be well-behaved enough for the error estimates to be useful. The CO data from the first three days after the event are not well fitted by the exponential used to fit the subsequent days, presumably because the plume has a small horizontal extent and lies in between the MLS orbit tracks, or because material is still being lofted into the height range where MLS can observe it. These days are therefore excluded from the exponential fit. The values of $M_0$ and $t_d$ obtained depend rather critically on exactly which of these early points are included. The given values are obtained using data from 16 August 2017 onward; this date is marked $t_0$ on Fig. 5 (a). This date was chosen because to use a later date is to throw away data that are part of the exponential decay, while to use earlier dates is to include data that are not part of the exponential decay. A cutoff date earlier than 16 August 2017 results in smaller estimates of $M_0$ and larger estimates of $t_d$; later dates give a range of similar values with larger errors. The values obtained are also sensitive to the end of the window used for the exponential fit, $t_e$, and to the full period used for the seasonal fit, between $t_1$ and $t_2$. A more conservative estimate of the parameters and their errors, taking into account the spread of values obtained with reasonable ranges of start and end dates, is $M_0 = 2450 \pm 210$ Gg and $t_d = 28 \pm 10$ days. We note that the decision to use a constant plus semiannual and annual cycles to fit the seasonal behaviour is an informed choice. Other reasonable choices could be made and would presumably result in similar errors.

To place the PNE in context we have applied this technique to three other significant fire events: the Australian new year (ANY) event of 2019-20 (Schwartz et al., 2020), the Black Saturday (BS) event of February 2009 (Pumphrey et al., 2011) and the Great Divide (GD) fire of December 2006 (Pumphrey et al., 2011). The results are shown in Table 1. We use a lower

maximum pressure for the BS event because the 215 hPa level showed very little CO from the event after the first few days. At the time it occurred, the PNE was a record-breaking event for the period of the MLS mission; it has since been superseded by the ANY event. We repeated the ANY calculation using only the 100 hPa level and higher altitudes. We obtained a mass estimate of about 1800 Gg in reasonable agreement with the $1500 \pm 900$ Gg above $\theta = 380$ K obtained by Khaykin et al. (2020).

Although the method described here is useful for CO and works very well for volcanic $SO_2$, it is not successful for the other

species described in this paper, mainly because the signal from the event is small compared to variability due to other causes. For the larger ANY event similar techniques can be used for $CH_3CN$ and $H_2O$; some results for these species are shown in Khaykin et al. (2020).

### 3.1.2  Water vapour

The event described in this paper differs from the Black Saturday event (Pumphrey et al., 2011) in that the polluted airmass has

a water vapour content which is clearly a great deal higher than in the surrounding atmosphere. For example, by the time the polluted airmass reaches 68 hPa it contains mixing ratios as high as 14 ppmv against a background of $3.5 - 5.5$ ppmv. Figure 6 shows the longitudes of unusually high water vapour in the same manner as Fig. 4. The values chosen for the parameter $\alpha$, described above for CO, vary from $\alpha = 6$ at 147 hPa to $\alpha = 2$ between 38 and 21 hPa. Many points with enhanced water vapour are found between 215 and 147 hPa and between $60°$ E and $180°$ E. These points occur in all years and are presumably caused

by the strong convection that occurs in this region in summer as an aspect of the Asian summer monsoon (see e.g. Jiang et al. (2010); Santee et al. (2017)). As they are not connected with the PNE we have removed from Fig. 6 all points between 215 and 147 hPa and between $60°$ E and $180°$ E. It is clear from a comparison between Figs. 4 and 6 that, where both species are enhanced, the enhanced CO and $H_2O$ occur in the same locations. The enhanced CO can be detected against the background variability by the method described 2–3 days before the enhanced $H_2O$ becomes apparent. Inspection of daily maps of the

data reveals that $H_2O$ is clearly enhanced on 14 August 2017 between $60°$N and $70°$N. The algorithm used fails to detect the enhancement at this time because of the very large background variability at lower latitudes.

After about 50 days the water vapour in the polluted airmass exceeds the background value to a greater extent than is the case for CO. After 63 days, the polluted airmass can only be identified in the water vapour data; this is presumably due to the short chemical lifetime of CO in the stratosphere and is consistent with the CO stratospheric lifetime in the ANY plume of

$\sim 30$ days estimated by Schwartz et al. (2020). A small parcel of enhanced $H_2O$ may be seen travelling westwards from $150°$ E to $80°$ W, between 65 and 90 days after the event, before ceasing to travel westwards and fading into the background. By this time, it is affecting the 31 and 26 hPa levels: an altitude of approximately 24 km.

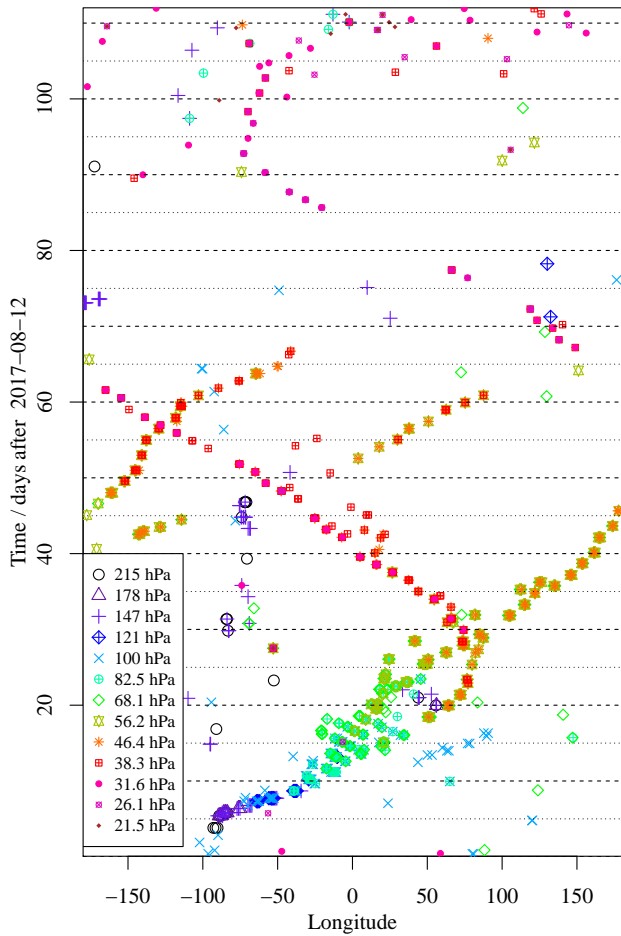

**Figure 6.** Longitudes of enhanced $H_2O$, plotted as a function of time. Colours and symbols represent pressure.

### 3.1.3 Other species

All chemical species observed by MLS were examined for the few months after the PNE using the same thresholding tech-
nique as for CO and $H_2O$. For ClO, we examined the standard (640 GHz) product, a secondary product derived from the
190 GHz measurements, and a product derived from the 640 GHz data while also estimating $CH_3OH$. The species fall into
three categories:

– Species for which no anomalous mixing ratios in the plume rise above the background variability: $O_3$, $SO_2$, BrO, $HNO_3$,
$N_2O$, ClO (190 GHz). Some of these species ($O_3$ and $N_2O$ are the most obvious) nevertheless exhibit some correlation
with CO values in the plume, as we show later.

– Genuine biomass burning products: CO, $H_2O$, HCN, $CH_3CN$, $CH_3OH$ and $CH_3Cl$; elevated values of these are observed
for about 60 days (100 days for $H_2O$).

**Table 2.** Summary of species observed by MLS to have clearly enhanced mixing ratios in the polluted airmass. The first and last days after 12 August 2017 are listed as $d_0$ and $d_1$; the highest and lowest pressures at which unusual values are observed are listed as $p_0$ and $p_1$. The "BB" column indicates whether or not the species is a genuine biomass burning product. For some species the enhanced values are thought to be caused by a measurement problem (see main text). Species observed by the instrument but with no unusual values detected in the polluted airmass are: $O_3$, $SO_2$, BrO, $HNO_3$, $N_2O$.

| Species | $d_0$ | $d_1$ | $p_0$ | $p_1$ | BB |
|---|---|---|---|---|---|
| CO | 2 | 63 | 316 | 31 | yes |
| $H_2O$ | 2 | 90 | 316 | 26 | yes |
| HCN | 8 | 59 | 100 | 31 | yes |
| $CH_3CN$ | 5 | 58 | 147 | 31 | yes |
| $CH_3Cl$ | 5 | 26 | 147 | 68 | yes |
| $CH_3OH$ | 5 | 16 | 147 | 68 | yes |
| ClO (640 GHz) | 5 | 33 | 147 | 46 | no |
| $HO_2$ | 5 | 11 | 147 | 68 | no |
| HOCl | 5 | 8 | 147 | 100 | no |
| HCl | 5 | 7 | 147 | 100 | no |

    – Species for which anomalous measurements are believed to be artefacts arising from spectral interference due to the presence of another molecule: HOCl, $HO_2$, HCl, ClO (640 GHz).

A summary of the ranges of time and altitude over which each species has unusual values is shown in Table 2. Pumphrey et al. (2011) demonstrate that an airmass containing biomass-burning products may appear from the MLS level 2 data to contain enhanced amounts of the 640 GHz ClO product, but that this is actually caused by enhanced amounts of methanol ($CH_3OH$), which has a similar spectral signature to ClO in the passband of the MLS 640 GHz radiometer.

    As is the case for many MLS data products, $CH_3OH$ is produced from a dedicated retrieval "phase", in which a ClO product

(not the standard one) is also retrieved (see Livesey et al. (2020) for details). The ClO data from the methanol phase show far less correlation with CO than the standard 640 GHz ClO product. This also suggests that the enhanced values seen in the standard 640 GHz ClO product are an artefact caused by enhanced methanol. HOCl, $HO_2$ and $CH_3Cl$ are all detected via spectral lines that are very close to the $CH_3OH$ lines. It seems reasonable to suppose that HOCl and $HO_2$ appear to be enhanced in the MLS data due to spectral contamination from $CH_3OH$, especially as they are both observed only for a very

short period. $CH_3Cl$, on the other hand, is known to be a biomass-burning product (Santee et al. (2013) and references therein); its spectral signature is sufficiently different from that of ClO (or $CH_3OH$) that the enhancement observed is almost certainly genuine. HCl, which shows the smallest enhancement in the plume, is observed at a different frequency from the ClO, $CH_3OH$ and $CH_3Cl$ lines, but one very close to the main $CH_3CN$ lines in the MLS 640 GHz radiometer. Although HCl is a minor fire product (mainly produced from the burning of garbage (Akagi et al., 2011)), it seems likely that the enhanced values observed

are spurious and are caused by the large amounts of $CH_3CN$ in the plume.

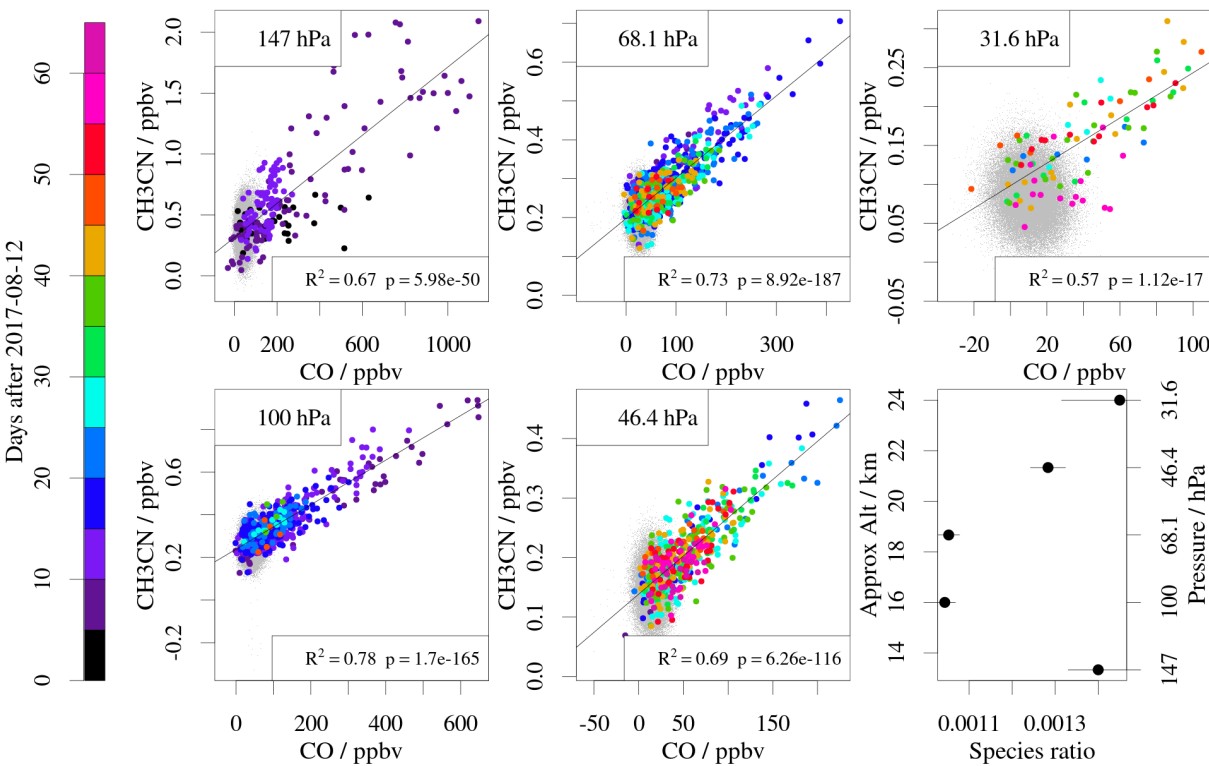

**Figure 7.** Scatter plots of the mixing ratio of CH₃CN against that of CO at five pressure levels between 14 August 2017 and 30 October 2017. Only data from north of $27.5°$ N are shown. The small grey dots show the entire dataset within the stated latitude and time ranges. The large coloured dots are points with enhanced CO and points within $350\,\mathrm{km}$ ($48\,\mathrm{s}$) of those points along the orbit track. The linear fits use the large points only and are performed using the `lm()` function of R (R Core Team, 2018); the $R^2$ and $p$-value are shown for each fit. The lower right panel shows the slopes of the linear fits as a function of altitude.

The relationships between species in the plume were investigated further by calculating a linear fit between enhanced CO and a given species at the same retrieval level and location. Data from profiles immediately adjacent to the plume are included to provide an appropriate baseline. We show an example in Fig. 7 in which the other species is CH₃CN. Plotting the data in this manner reveals that in some cases ($O_3$, $N_2O$) there is a correlation between the two species, even when the non-CO species does not have values outside its usual range. For other species ($SO_2$, $HNO_3$, BrO), no such correlation is observed, or the correlation is small enough to suggest that it is an instrumental rather than a geophysical effect. We note values of the linear fit slope in Table 3 and show a selection of the data from Table 3 in Fig. 8. For some species (including the CH₃CN shown in Fig. 7) we have used data outside the vertical range recommended by the MLS team. This was done where the correlation with CO was clear; we note that the linear fit slopes will not be affected by a constant bias in the mixing ratio of either CO or the other species. For species produced by biomass burning and that have similar lifetimes to CO in the lower stratosphere, we would expect the ratios observed in Table 3 to be constant with altitude and to be in reasonable agreement with the emission

**Table 3.** Ratios between the mixing ratio of various species and that of CO, at various pressure levels. Subscripts indicate errors in the last digit. The errors are those returned by a standard fitting program (the `lm()` function in R). The right-hand column shows values from the literature converted into ratios with CO. Superscripts in this column indicate sources: Randerson et al. (2017)[1], Rinsland et al. (2007)[2], Simpson et al. (2011)[3], Akagi et al. (2011)[4]. Published values shown are all for boreal forests with the exception of HCl. A "—" indicates either that the species does not have good data at that level or that the linear fit is not significant at the 99% level ($p > 0.01$).

| Species | 215 hPa | 147 hPa | 100 hPa | 68 hPa | 46 hPa | 31 hPa | Other measurements |
|---|---|---|---|---|---|---|---|
| $H_2O$ | $119_5$ | $47_2$ | $18.5_8$ | $32.2_7$ | $39_1$ | $35_4$ | — |
| HCN | — | — | — | $0.00097_4$ | $0.00167_4$ | $0.0022_2$ | $0.012^{1,4}$, $0.00242^2_{32}$, $0.0082^3_{20}$ |
| $CH_3CN$ | — | $0.00140_7$ | $0.00104_3$ | $0.00105_3$ | $0.00128_4$ | $0.00145_{14}$ | $0.0032^4$, $0.0018^3_3$ |
| $CH_3Cl$ | — | $0.00109_7$ | $0.00141_5$ | $0.00127_7$ | $0.0012_1$ | $0.0011_4$ | $0.0056^2$, $0.00014^3_3$ |
| ClO std | — | $0.0048_2$ | $0.0056_2$ | $0.0044_1$ | $0.0036_2$ | $0.0014_5$ | — |
| HCl std | — | $0.0039_2$ | $0.0018_1$ | $-0.0010_2$ | $-0.0033_4$ | — | $0.0010^4_{10}$ (chaparral) |
| $CH_3OH$ | — | $0.012_1$ | $0.0107_5$ | $0.0058_2$ | $0.0062_4$ | $0.0024_9$ | $0.019^{1,4}_{13}$, $0.00015^3_{15}$ |
| $O_3$ | $-0.15_2$ | $-0.23_3$ | $-0.39_8$ | $-1.81_{15}$ | $-5.2_3$ | $-6_1$ | — |
| $N_2O$ | — | — | — | $0.21_1$ | $0.32_3$ | $0.4_1$ | $0.0020^1$ |

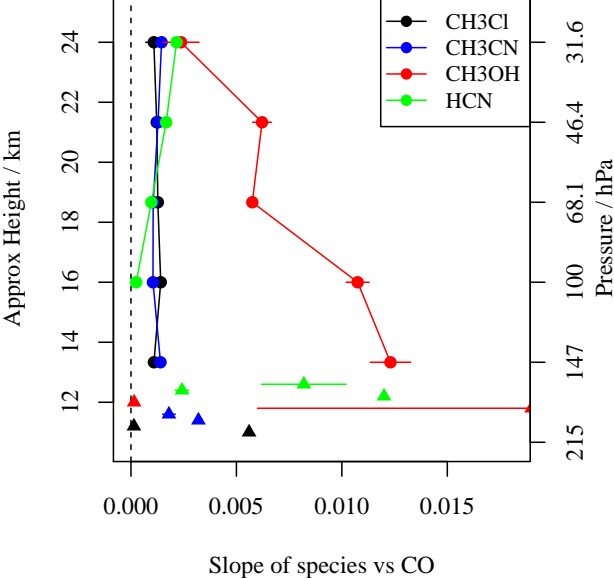

**Figure 8.** Slopes of scatter plots of various species against CO; the slopes are calculated as in the example shown in Fig. 7. Triangles are the data from the "other measurements" column of Table 3, with their error bars (where available). All the triangles refer to measurements made in the troposphere; their position in the vertical has no meaning.

ratios measured previously. The relationships between CO and CH$_3$CN, and between CO and CH$_3$Cl show relatively little variation with height. The slopes we obtain are generally within the published ranges; for CH$_3$Cl the range is rather large. The methanol/CO ratio is within the rather wide range of published values and decreases with height. This might appear to suggest that methanol undergoes chemical loss in the plume more rapidly than CO, but the MLS methanol data are potentially useful only at 147 hPa (and at low latitudes at 100 hPa), so any such conclusion must be regarded as speculative. It would nevertheless be consistent with the $\sim 10$ day lifetime for methanol estimated in Schwartz et al. (2020). The agreement between our value for the methanol/CO ratio at 147 hPa and the published values is reasonable given the large variability in the latter.

The H$_2$O/CO ratio varies considerably with height between 215 and 68 hPa, but is approximately constant between 68 and 31 hPa. The standard 640 GHz ClO data behave in a similar manner to the biomass-burning products. This is compatible with the fact that the standard ClO retrieval may have been affected by interference or contamination from methanol (Pumphrey et al., 2011); as noted above, the other ClO products are not enhanced so the elevated abundances of the standard product are probably due to high values of methanol.

Yu et al. (2019) showed that the negative anomaly in ozone and the positive anomaly in H$_2$O observed by MLS were a consequence of the transport of tropospheric air into the stratosphere (and not in situ chemical loss in the case of ozone). Such upward transport probably also gave rise to increases in N$_2$O and thus positive correlations between N$_2$O and CO. Kablick et al. (2020) discuss the low ozone and high N$_2$O values associated with the ANY fires of December 2019. They attribute these anomalies to rapid ascent of tropospheric air; it appears likely that the ozone and N$_2$O in the PNE behaved in a similar way. Khaykin et al. (2020), on the other hand, suggest that the low ozone values observed in the ANY event are at least partly due to chemical depletion.

## 3.2 Trajectory and dispersion modelling

### 3.2.1 Back Trajectories

Back trajectories from the MLS observations in the first few days after the event were calculated using the FLEXTRA trajectory model (Stohl et al., 1995), driven by analyses from the National Centers for Environmental Prediction (NCEP) Global Forecast System (GFS) (National Centers for Environmental Prediction, 2000). FLEXTRA has no mechanism for modelling the rapid rise of the polluted airmass; for this reason we restrict ourselves to trajectories started less than four days after the fire. Trajectories started at 316 and 215 hPa pass reasonably close to the Pacific Northwest Event; we show examples in Fig. 9. Although MLS observes enhanced CO at the 147 hPa level at this time, trajectories started at this level pass far to the west of the PNE. The vertical resolution of the MLS grid is six levels per pressure decade, or about 2.7 km, and the true vertical resolution at the levels of concern here is 4.5 − 5 km (Livesey et al., 2020). If the enhanced CO were in a thin layer at an altitude between two of the MLS reporting levels, then enhanced CO would be reported at both of those levels. The fact that trajectories started at 147 hPa do not pass close to the fire location, and that trajectories started at 215 hPa do pass close to the fire, suggests that over these first few days the polluted airmass is closer to 215 hPa than to 147 hPa. It is also possible that the polluted air extends to altitudes lower than 215 hPa, but we cannot infer this reliably as the MLS data at 316 hPa have a larger

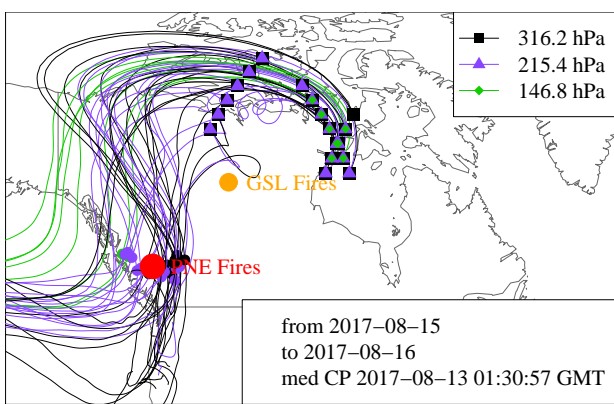

**Figure 9.** Back trajectories started at points (non-circular symbols) where MLS observed large values of CO. Trajectory colours distinguish different pressures at the starting point; the trajectories are three-dimensional. The location of the Pacific Northwest Event (PNE) is marked in red; that of the Great Slave Lake (GSL) fires in orange. Where the trajectories pass within 400 km of the PNE, the closest point is marked with a circular dot.

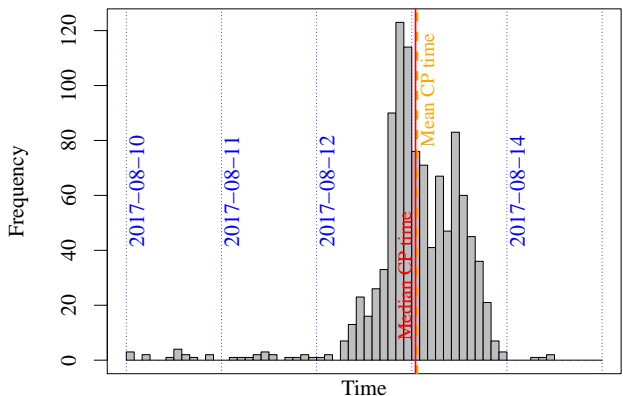

**Figure 10.** Histogram of closest-pass times of trajectories to the PNE. The median and mean values are marked in red and orange respectively. The dotted blue lines are at 00:00 UTC.

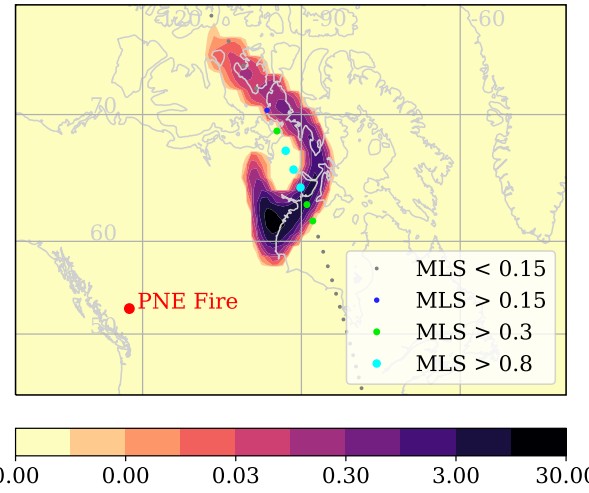

**Figure 11.** FLEXPART CO mixing ratio in ppmv at an altitude of 12 km on 15 August 2017 at 18:00. Dots show the location of MLS profiles from an orbit at a matching time. The dots are coloured to show where MLS CO, interpolated to an altitude of 12 km, is below 0.15 ppmv, and where it exceeds 0.15, 0.3 and 0.8 ppmv.

contribution from 215 hPa than from 316 hPa (Livesey et al., 2020). There were other forest fires burning at the same time as the PNE, notably in the region around Lake Athabasca and the Great Slave Lake (GSL); the location of these fires is marked on Fig. 9. The trajectories make it reasonably clear that the source of the polluted airmass observed by MLS is the Pacific Northwest Event and not the fires near the Great Slave Lake, a point already noted by Fromm (2019).

By starting trajectories at many points spread between the MLS observations of high CO, both vertically and along the
measurement track, we obtain sufficient times of closest pass to the PNE to construct a histogram, as shown in Fig. 10. The mean and median closest pass times are 01:20 UTC and 00:57 UTC respectively on 2017-08-13. The standard deviation of the closest-pass times is 12 hours. The closest pass times are in good agreement (given their own standard deviation) with the times reported by Peterson et al. (2018), who state that pyrocumulonimbus clouds (pyroCb) are observed to develop between 23:00 UTC on 12 August 2017 and 00:30 UTC on 13 August 2017, and that they remain active for 1–4 hours.

**3.2.2    Plume trajectory modelling**

The FLEXPART model (Stohl et al., 2005) was used to model the dispersion of the plume from the Pacific Northwest Event. FLEXPART allows the user to release a mass of a pollutant at a chosen place and time and to observe how it disperses. We released the pollutant in a small cuboid volume. The release is defined by the location and size of this volume in latitude, longitude and altitude, by a start and end time, and by the mass released: nine parameters in total. We show in Fig. 11 an
example of FLEXPART output for a time approximately 66 hours after the PNE. An MLS orbit that passes through the polluted airmass at that time is also shown; points where the MLS CO is enhanced are marked and lie close to the modelled plume.

We adjust the properties of the mass released to best match the MLS observations using the Markov chain Monte Carlo (MCMC) inverse modelling technique (van Ravenzwaaij et al., 2018). As FLEXPART is not able to model the self-lofting of the polluted airmass, we used only the first few days of data after the fire; the last data used are from 18 August. We also restrict the data used to those poleward of 40°N and between 40°W and 140°W. FLEXPART output is on altitude surfaces. In order to compare it with the MLS data we ran FLEXPART so that its output was in mixing ratio units on a vertical grid with a 1 km spacing. The MLS data were interpolated onto this vertical grid from their native pressure grid using the MLS geopotential height data. The FLEXPART output was interpolated to the MLS locations in order to calculate the differences. FLEXPART is set up to assume that the atmosphere contains no CO before the time of the fire; to match this the MLS data are converted to anomalies by subtracting a zonal average value for the few days before the event. The MCMC technique used is the basic Metropolis-Hastings algorithm. It starts with a reasonable guess at the parameters required to define the release of CO. It then makes a sequence of small alterations to the parameters, accepting them if they make the FLEXPART output agree better with the MLS data. Parameters that agree less well than the previous set are accepted with a probability that depends on how much worse the fit is. The resulting collection of sets of parameters should, if the process is continued long enough, have a probability distribution reflecting the measurement and modelling errors. To ensure that the results are not dependent on the starting point chosen, several chains with different starting points were run. The "burn-in" period for each chain was rejected by inspection.

Initial attempts to estimate all nine release parameters by this method demonstrated that the data do not provide sufficient information to determine the horizontal extent of the release region, nor its extent in time. We therefore constrained the region to be 0.5° wide in longitude, 0.375° wide in latitude and 1.5 hours in duration. These choices were intended to ensure that the box was similar in size to the observed pyroCb clouds. The six estimated quantities were then the latitude, longitude and time of the release cuboid, its vertical extent, and the total mass of CO released.

We show the horizontal release locations deemed acceptable by the MCMC approach in Fig. 12; each of the coloured dots represents the centre of a release. The size and shape of the release box is also shown. The average location of the acceptable releases is 53.2°N, 121.8°W. There were several large anvil clouds in the area at the time; these are clearly visible in AVHRR images and are outlined in green in Fig. 12. The MCMC solutions have a standard deviation of about 6 km perpendicular to the wind direction and 30 km in the wind direction. The injection altitude is determined rather precisely: $11.5 \pm 0.1$ km; as expected from the back trajectories, this is closer to 215 hPa than to 147 hPa. The injected mass is $620 \pm 80$ Gg, considerably smaller than the value of $2450 \pm 210$ Gg that we obtained by the simple method discussed in section 3.1.1. This is qualitatively reasonable, given that the MCMC approach only considers CO that is observable by MLS near the start of the event, while the simple method of section 3.1.1 could include CO that was injected at too low an altitude for MLS to observe it and which subsequently ascended to observable altitudes. We consider that the method of section 3.1.1 gives the more reliable mass estimate not only for this reason, but also because it has given mass estimates of $SO_2$ from volcanoes that are in broad agreement with a variety of independent measurements (Pumphrey et al., 2015). Although the MCMC method does not make use of data from as many days as the simple method of section 3.1.1, it is worth noting that it uses all the MLS data within the latitude and longitude ranges noted above. The release parameters are constrained not only by the requirement to show enhanced CO where MLS observes it, but also to show no enhancement where MLS observes none.

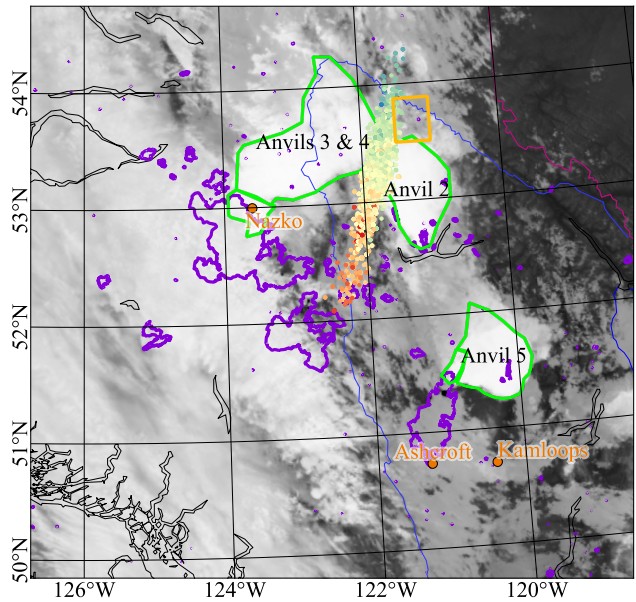

**Figure 12.** Map of the source region, showing the areas that burned in 2017 (purple; some areas are sufficiently small that the contours enclosing them appear as dots) and anvils observed in AVHRR data (green). The anvils are numbered using the same scheme as Peterson et al. (2018); their anvil 1 is outside the area shown here. The grey-scale backdrop is the AVHRR channel 5 (thermal IR) image taken at about 02:00 GMT. The orange rectangle shows the size and shape in the horizontal of the cuboid release volume. The small points are the MCMC solutions for the apparent origin of the polluted airmass and are coloured according to time; varying from 2:00 GMT on 2017-08-13 (red) to 7:30 GMT (blue).

The average release time is 5:20 UTC on 13 August 2017. The acceptable solutions range from 2:00 UTC to 7:30 UTC and have a standard deviation of 40 minutes. This is later than the time estimated using FLEXTRA trajectories, but is well within the 12 hour standard deviation of the FLEXTRA results. As noted above, Peterson et al. (2018) report the pyroCb events as 285 beginning between 12 August 2017, 23:00 and 13 August 2017, 00:30 and lasting 1-4 hours. The release times we determined using FLEXPART and the MLS data are at the late end of this range.

## 4  Discussion

Significant injections of pyroconvective plumes into the stratosphere generally require both extremely intense fires and particular meteorological conditions, such as those preceding the passage of a cold front (Peterson et al., 2018). The meteorology 290 surrounding the Pacific Northwest Event is described in some detail by Peterson et al. (2018); the fire indeed occurred while a cold front was approaching.

Most forest fires do not produce a pyroCb cloud. Moreover, most pyroCb clouds do not extend to a great enough altitude to loft the products high enough for MLS to observe them. MLS has observed no other events comparable to the PNE in North

America during a 17-year mission. According to an incomplete accounting of pyroCb events Peterson et al. (2017) there are very roughly 20–35 pyroCbs per year in western North America. This number of pyroCbs is itself small compared to the total number of boreal fires. The Government of British Columbia (2020) report an annual average of 1356 distinct fires over the last decade, while Stocks et al. (2003) report on the order of 8000 large ( > 200 ha) fires per year across the whole of Canada.

The only events in the MLS record that are in any way comparable to the PNE are the Black Saturday (BS) fire, which occurred in Australia in February 2009 (Pumphrey et al., 2011), and the ANY fires of December 2019 to January 2020 (e.g. Khaykin et al. (2020); Kablick et al. (2020); Schwartz et al. (2020)). The mass of CO injected into the lower stratosphere by the BS fire was, as we show in section 3.1.1, 1350 Gg — just over half of the mass injected by the PNE. Like the PNE, the Black Saturday fire occurred at a time when a cold front was approaching (Dowdy et al., 2017). The 2019/20 ANY fire injected a mass of CO considerably larger even than the PNE: initial estimates, also shown in section 3.1.1, give a mass of 7700 Gg. MLS also observed enhanced CO from the Great Divide fires in Australia in December 2006. These fires burned for two months, but MLS only observed enhanced CO from them for a short time, beginning on 12 December 2006. The injected mass of CO from the Great Divide fires was 1520 Gg, similar to that from BS, but the polluted airmass did not ascend in the manner observed with the PNE, ANY and BS events, and was observed by MLS for only 10 days (Pumphrey et al., 2011).

The MLS data shown in this paper are for most species the first data reported from the PNE. Measurements of aerosols emitted by the event have been reported in some detail. Ansmann et al. (2018), Khaykin et al. (2018), Zuev et al. (2019) and Hu et al. (2019) report ground-based lidar observations of the PNE plume as it passed over Europe. Khaykin et al. (2018) also report aerosol data from the CALIOP lidar instrument on the CALIPSO satellite. The maps of the CALIOP data are similar to Fig. 3, indicating that the aerosol and CO travelled together; presumably the aerosol particles were too small to settle out on the timescale over which the CO was observed. Khaykin et al. (2018) suggest that the pollution is from fires near Lake Athabasca; we show clearly that it is from British Columbia and from the vicinity of the Pacific Northwest Event as described by Peterson et al. (2018). Ansmann et al. (2018) ascribe the source of the pollution to British Columbia in agreement with our conclusions. Lestrelin et al. (2021) track the event in detail using a combination of CALIOP lidar data and ERA5 reanalyses. Their description of the event agrees in many details with our own analysis based on the MLS $H_2O$ and CO data. Kloss et al. (2019) analyse the event using aerosol data from the Stratospheric Aerosol and Gas Experiment III on the International Space Station (SAGE III/ISS, NASA/LARC/SD/ASDC (2017)) and the Ozone Mapping and Profiler Suite Limb Profiler (OMPS-LP, Loughman et al. (2018)) instruments, claiming that the aerosol is transported around the eastern edge of the Asian monsoon anticyclone (AMA) into tropical latitudes. We observe no such behaviour in the MLS CO data. One reason for this is that by the time parts A and B1 (as labelled in Fig. 4) of the polluted airmass are in the vicinity of the AMA, they are at altitudes higher than the AMA and also higher than the aerosols discussed by Kloss et al. (2019).

In addition to the MCMC results, Fig. 12 shows the areas that burned in 2017 marked in purple (British Columbia Wildfire Service, 2018). Figure 12 also shows the AVHRR thermal image; three large anvils of high cloud are clearly visible. Examination of the visible bands of the same image suggests that the anvils marked "Anvil 3 & 4" and "Anvil 5" have an active pyroCb cloud at the southwest edge. It is not clear from the visible images that an active pyroCb is associated with Anvil 2; images

taken during the previous two hours (Peterson et al., 2018) show that Anvil 2 formed near 00:00 UTC. Anvil 3 also formed around 00:00 UTC; Anvil 4 formed half an hour later and had merged with Anvil 3 by 01:45 UTC.

330 The possible solutions for the source of the polluted airmass that emerge from the MCMC inverse method are shown in Fig. 12 as a cloud of small points. The colours of these dots indicate the time, varying from 2:00 GMT (red) to 7:30 GMT (blue). The location of these dots suggests that the polluted airmass observed by MLS is probably associated with Anvils 2, 3 and 4 and is less likely to be associated with Anvil 5. This in turn suggests that the emissions come from the Plateau complex of fires (west of Nazko) and not from the Elephant Hill fires (north of Ashcroft). It could be argued that the polluted airmass

335 did not come from the region shown in Fig. 12, but rather was injected at some point along the trajectory between this region and the first MLS observation of enhanced CO. We consider such a possibility unlikely as the injection would have to have occurred in a manner that matched the shape of the polluted region as modelled by FLEXPART at the time of the hypothesised injection.

 The polluted airmasses from the PNE and ANY contained large amounts of water — this distinguishes them from the Black

340 Saturday event, in which the water content of the polluted airmass was similar to that of the surrounding air, except at 100 hPa and lower altitudes during the first few days of the event.

 Many studies (e.g. Williams and Abatzoglou (2016)) suggest that climate change over the coming decades is likely to lead to an increase in forest fires. However, events like the PNE are rare, are a very small fraction of the total number of forest fires and require conditions in addition to the presence of a fire. Therefore, we do not consider that we can predict whether events

345 similar to the PNE will become more common in the future.

 The rapid ascent of the polluted airmass from the PNE after its initial injection results from solar heating of its extraordinarily dense black carbon aerosol (Yu et al., 2019; Kablick et al., 2020). The event can in some ways be considered a small-scale demonstration of what would happen in the aftermath of a nuclear war: such a war would cause widespread fires that would put a great deal more black carbon into the middle atmosphere than the PNE did. Self-lofting of this carbon (as demonstrated

350 by the PNE) would raise it to an altitude where it would remain for a number of years, giving rise to the so-called *nuclear winter* (Robock et al. (2007) and references therein). Simulations of the response of the climate to a regional-scale nuclear war disagree on the severity of this effect (Robock et al., 2007; Reisner et al., 2018). However, much of the disagreement is caused by widely different estimates of how much black carbon would reach the stratosphere from a given nuclear explosion. Once it did reach the stratosphere, the black carbon aerosol from a nuclear war would ascend in the manner that we observe in the

355 PNE.

## 5 Conclusions

The Pacific Northwest Event was a forest fire accompanied by unusually strong pyroconvection, injecting both aerosol and gas-phase pollutants into the stratosphere. Observations of CO from the MLS instrument show that the polluted airmass reached all longitudes on a timescale of 60 days and ascended to a pressure of 46 hPa (approximately 22 km altitude). The polluted

360 airmass was much wetter than the surrounding air in the stratosphere; in this respect it was different from the otherwise similar

Black Saturday event and similar to the recent ANY event. In water vapour, the plume was observable for 100 days, ascending to 31 hPa (approximately 24 km altitude).

The polluted airmass contained biomass-burning products HCN, $CH_3CN$, $CH_3Cl$ and $CH_3OH$ in broadly similar proportions to CO as observed by other techniques in other biomass-burning events. The evolution of the gas-phase components of the polluted airmass as observed by MLS was similar to that of its aerosol components reported elsewhere (Khaykin et al., 2018; Lestrelin et al., 2021).

Back trajectories link the observed excess CO clearly to the PNE as described by Peterson et al. (2018), and more detailed plume dispersion modelling shows that the pollution observed by MLS probably came from the Plateau complex of fires to the west of Nazko. The time of injection given by this method is in agreement with the description of the event given by Peterson et al. (2018).

A simple approach to estimating the mass of CO injected above 215 hPa leads to a value of 2450$\pm$210 Gg. We consider this number more reliable than the lower number of 620$\pm$80 Gg obtained by plume dispersion modelling. However, we also consider that the discrepancy bears further investigation.

Events of the type we describe here are rare; the PNE, ANY, Black Saturday and Great Divide fires are the only such events in the MLS dataset at the time of writing. Whether they will become more or less frequent as the climate changes over the coming decades remains to be seen.

*Code and data availability.* The FLEXTRA and FLEXPART models are available at https://www.flexpart.eu/. The MLS data used in this paper are available at https://disc.gsfc.nasa.gov/. The data for each species has a separate DOI, e.g. `10.5067/Aura/MLS/DATA2005` (CO), `10.5067/Aura/MLS/DATA2009` (H2O). The AVHRR L1B data can be obtained from https://www.bou.class.noaa.gov/saa/products/welcome. The GFS analyses (National Centers for Environmental Prediction, 2000) are available from https://doi.org/10.1038/s41612-018-0039-3.

*Author contributions.* HCP wrote most of the paper and carried out a large part of the data analysis and modelling. MJS provided some of the data analysis and suggested the use of figures 4, 6 and 7. MLS provided advice on the $CH_3OH$, ClO, $CH_3CN$ and $CH_3Cl$ products, was responsible for the use of Fig. 8 and suggested many improvements to Fig. 7. MDF assisted with the trajectory analysis. All co-authors commented on multiple drafts prior to submission, helped to shape the analysis, provided HCP with a great deal of assistance in responding to the referees' comments and suggested many improvements to the text.

*Competing interests.* The authors declare that they have no competing interests.

*Acknowledgements.* The authors thank Albert Ansmann for drawing our attention to two important references. They also thank both referees and the editor for their diligence in reviewing and editing the paper. Work at the Jet Propulsion Laboratory, California Institute of Technology,

was done under contract with the National Aeronautics and Space Administration. MLS research at Edinburgh was funded by NERC under grant NE/E003990/1 and previous grants.

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
