# Peer review of "MLS observations of biomass-burning products in the stratosphere from Canadian forest fires in August 2017"

_Atmospheric Chemistry and Physics, 2020_

## Referee Comment (RC1) · Anonymous Referee #1 · 4 Sep 2020

The paper details multiple analyses of the lower to middle stratospheric after the Pacific Northwest Event biomass burning event in August 2017. Using data from Aura MLS, the polluted airmass is tracked around the world in the months after the injection. The authors analyze the composition of the polluted airmass and attempt to track it to its specific source. Overall, the work is sound, though more elaboration on the analyses and results are needed in various locations in the paper and the authors should provide more quantitative results on species other than CO.

**General Comments**
In Section 3.1 (Pg. 02, Lns. 044–51) the authors describe a technique (used previously in Pumphrey et al. (2011)) to distinguish enhanced amounts of a gas species versus

background levels. This scheme uses the mean and standard deviation in an iterative fashion to flag values that stand apart from the main distribution. This appears to be somewhat of a convoluted (and admittedly subjective) process. Have the authors thought about using simple statistical metrics more apt at detecting outliers such as the median and median absolute deviation, or is the frequency of enhancement so high as to be bimodal (in which case a completely different method should be performed)?

Are the data used in Figure 2 poleward of 25N or 15N? The caption appears to have a typo.

I assume the oscillatory appearance of enhanced CO in Figure 3 over North America is a byproduct of the MLS sampling (and not any transport behavior), but that it does reveal the meridional extent of the plume. If true, it might be worth mentioning in the figure caption as a clarification.

What is the semi-transparent red circle at roughly 70N and 125W in Figure 3?

Pg. 04, Ln. 059: "... the polluted airmass is divided into two parts on three occasions: at approximately 9 days, 16 days and 41 days after 12 August 2017"
What about the split at 30 days?

I was wondering what the causes of the different splits could be and whether they might play a role in the correlation between CO and H2O. For example, the split around 9 days occurs over the Atlantic and appears to be associated with some lifting. Hurricane / Tropical Storm Gert was active around that time and location, is it possible that it played a role in this split and ascent? Similarly, the split around 30 days occurs at the border of the Tibetan plateau during the Asian Monsoon (the eastward path

goes around the northern side while the westward path goes south above the plateau towards lower latitudes and is also associated with some lifting).

The split at 41 days is interesting because it is difficult to tell if it really is a split. Trying to follow along between Figs. 3 and 4, the paths are disjointed. It does appear that the airmass moving East over northern Japan becomes the so-called "slower part" in Fig. 4. However, the "faster part" appears about 15 degrees North of this airmass and then subsequently moves over Alaska, Northern Canada, Quebec, before reappearing over Europe. Could the disconnect be an observation gap in MLS observations or the technique for determining "enhancement" not being sensitive enough at that time? Could the split have anything to do with Typhoon Talim that was in that area at that time (and also possibly affecting the amount of H2O)? Could it possibly not have been a split and instead been new material injected at that time, which might explain the small bump/deviation in Figure 5b around late September? Does a plot similar to Fig. 4, but using latitude instead of longitude, help visualize this at all?

Figure 5 says it shows all data poleward of 27.5N. Is there a reason this is different from Fig. 2?

Pg. 05, Ln. 068: "The seasonal behaviour of CO is fitted with a mean value and annual and semiannual sinusoids as shown in Fig. 5(a); this fit ignores the data for a period of 99 days after the PNE."
Over what total date range is this fit to the seasonal cycle determined?

Pg. 05, Ln. 071: "... we fit the decay with an exponential ..."
While often popular, the decay is almost never linear in log space and allowing for some degree of curvature would be more precise. Unfortunately, this makes a simple

calculation of the decay constant impossible and likely contributes to the wide range of possibilities.

Pg. 05, Ln. 075: "... presumably because the plume has a small horizontal extent and lies in between the MLS orbit tracks."
Or was material was still being injected / lifted?

Pg. 06, Ln. 088: "These points are clearly not connected with the PNE so we have removed them from Fig. 6."
What were the criteria for separation and what are the origins of these removed data? Also, the authors state that these are removed from Fig. 6, but it definitely still appears to contain data that likely have nothing to do with the polluted airmass.

Pg. 07, Ln. 093: "The algorithm used fails to detect the enhancement at this time because of the very large background variability at lower latitudes."
Would a different detection method such as the median and median absolute deviation help with this?

Pg. 07, Ln. 095: "After about 50 days the water vapour in the polluted airmass exceeds the background value to a greater extent than is the case for CO. After 63 days, the polluted airmass can only be identified in the water vapour data; this is presumably due to the short chemical lifetime of CO in the stratosphere."
Might a figure showing a time-series of H2O like Fig. 5 be useful here?

If HCl is an product that is considered enhanced as an artifact, then why is it included in Figure 8 (and the only one of those products included)?

The authors state that the computed slopes of correlation are within the expected ranges. Can the authors show the expected ranges either in Table 2 or Figure 8?

Pg. 12, Ln. 161: "This suggests that over these first few days the polluted airmass is closer to 215 hPa than to 147 hPa."
Can the vertical resolution of MLS be playing a role here? Also, this might suggest that the airmass was rising in a way that the model was not properly accounting for (e.g., self-lofting of the black carbon in the plume).

Pg. 13, Ln. 190: "The injected mass is 620±80Gg, considerably smaller than the value of 2400±300Gg that we obtained by the simple method discussed in section 3.1. This is qualitatively reasonable, given that the MCMC approach only considers CO that is observable by MLS near the start of the event, while the simple method of section 3.1 includes CO that was injected at too low an altitude for MLS to observe it and subsequently ascended to observable altitudes."
I would assume that some of this discrepancy is also likely because the initial MLS measurements do not cover the full horizontal extent of the spread of the polluted airmass at this time. I believe this is what the authors are trying to say but it could be made more explicit. Additionally, regarding the comment of altitudes of injection, do the MLS observations not reach the tropopause?

Pg. 14, Ln. 207: "The mass of CO injected into the lower stratosphere by the Black Saturday fire was, when estimated by the technique of section 3.1, 1.3 Tg – just over half of the mass injected by the PNE."
Is this result from an analysis first performed in this work or does it derive from another source? If the latter, it should be referenced. If the former, then more elaboration on the analysis needs to be included than simply the result.

Pg. 14, Ln. 209: "The 2019/20 ANY fire injected a mass of CO considerably larger even than the PNE: initial esimates **[there's a typo here]**, again using the method of section 3.1, place this mass in the range 8–10 Tg."
In the same line as my previous comment, any additional analysis needs to elaborate more on at least the temporal and spatial extent of the data used. Also, while I understand the desire to compare the PNE to the more recent and even larger ANY event, the authors already claim to have a paper in preparation so I find it quite unusual to throw in preliminary results of that analysis in this paper.

Pg. 15, Ln. 232: "The location of these dots suggests that the polluted airmass observed by MLS is probably associated with Anvils 2, 3 and 4 and is less likely to be associated with Anvil 5. This in turn suggests that the emissions come from the plateau complex of fires (west of Nazko) and not from the Elephant Hill fires (north of Ashcroft)."
The analysis performed here is with MLS data that is highly localized. The analysis supports the stated origin of the polluted airmass observed by MLS, but it does not rule out the other locations as potential sources that injected polluted air into regions not immediately observed by MLS.

The entire discussion about comparing black carbon and nuclear war on page 15 (lines 240 to 247) seems very out of place in this paper.

Pg. 15, Ln. 251: "The polluted airmass was much wetter than the surrounding air in the stratosphere . . ."
How much wetter? There do not appear to be any quantitative results of H2O in this paper.

Why does the paper discuss other biomass burning products (i.e., HCN, CH3CN, CH3OH, and CH3Cl) and perform correlation analyses and yet only computes an injection mass for CO? This should be performed or the title changed to something centered on CO.

It would be good to include some more results (both qualitative and quantitative) in the conclusion.

---

## Referee Comment (RC2) · Pasquale Sellitto (Referee) · 2 Nov 2020

The manuscript "Stratospheric pollution from Canadian forest fires" shows satellite (basically with the MLS sensor) observations of different gaseous pollutants in the smoke plume derived from the well-studied Canadian wildfire event in 2017, as well as some complementary studies with back-trajectories to determine the specific plume source at the surface. This work looks like a complement to the previous paper by Yu et al., 2019. The topic is with no doubt of interest for the ACP readers. While I see some potential in this manuscript, I cannot recommend its publication in the present form due to the following major and specific issues:

Major comments:

[Figure]

1)The originality of the manuscript must be clarified and openly discussed, in particular with respect to the previous work of Yu et al., 2019. What's new and what's complementary with respect to Yu et al., 2019? This is not clear at all in the present manuscript version, at least to me.

2) The paper is unfortunately very confusing and certainly not very well crafted. The Abstract, Introduction and Conclusions are very short, incomplete, probably rushed. Many statements are not justified but just "written down". Many discussion and argumentations are just lacking. All the analyses with trajectories (Sect. 4.2) are very obscure and must be extensively clarified/rewritten. Many crucial references are missing. The Authors should put a much larger effort in the writing of the text, the production of intelligible figures and the discussion of their results in the context of previous literature.

Details on these major comments and more specific comments are given in the following:

1) Title: The title looks like a bit too general. "Pollution" with respect to what? With which tools? Please be more specific.

2) Abstract: I feel that the Abstract is totally inadequate to represent concisely the overall scopes of the study and the results. Please rewrite it in a more extended way so to include scopes and results of the manuscript.

3) Introduction: The section called "Introduction" is not nearly sufficient to introduce the present work. This must be significantly widened to frame the present study in its general context. Examples (not exhaustive): Why we study fires? What are their impacts on the atmosphere (perturbations with respect to background conditions)? How we study fires? What we know about British Columbia fires so far? Please extend significantly the Introduction so that this section is a real "Introduction" to the present work.

4) L10-11: "On 12August. . .(215 hPa)": Please add a reference.

5) L11: "The fires are described in some detail...": What do you mean with "some" detail?

6) L12-14: ''The polluted airmass...(26 hPa)": All this is described by Peterson et al.? Please clarify.

7) L16-17: In Khaykin et al. there is a very detailed description of the event, so please develop what you mean with "the global evolution of the plume".

8) L16: Just a suggestion but "LiDAR" is an acronym (I prefer "LiDAR" rather than "lidar").

9) Introduction: About the BC fire plume dispersion, the Authors are missing here an important reference about the larger scale dispersion of the plume: Kloss, C., Berthet, G., Sellitto, P., Ploeger, F., Bucci, S., Khaykin, S., Jégou, F., Taha, G., Thomason, L. W., Barret, B., Le Flochmoen, E., von Hobe, M., Bossolasco, A., Bègue, N., and Legras, B.: Transport of the 2017 Canadian wildfire plume to the tropics via the Asian monsoon circulation, Atmos. Chem. Phys., 19, 13547–13567, https://doi.org/10.5194/acp-19-13547-2019, 2019. Please add the reference and a discussion, in particular in terms of the larger scale dispersion of the BC fires plume.

10) Introduction: A detailed description of: a. the results of Yu et al., 2019; b. what is new in the present work, with respect to Yu et al., and worth publication is required in the Introduction.

11) Sect. 3: A general section title "Results" (joining "Observations", "Trajectories" etc) seems necessary here

12) L34: "recommended": Recommended by whom?

13) L34-37: Please justify these statements.

14) L40-41: "this is about...value": Where are these values (zonal mean; daily maximum) taken from?

15) Fig. 1: Please put units next to the colorbar

16) L55-56: "...and the other returning westwards across the Atlantic at about 33 N.": this is probably linked to the Asian Monsoon anticyclone circulation - see Kloss et al., 2019, mentioned in a previous comment, please add discussion.

17) Fig. 5: "PNE" not visible in panel a. Also, please put more human-readable dates in the xaxis.

18) L69 and Fig. 5: Where is the annual sinusoid? I see only a semiannual one. Please be more precise in the text about periods for the fitting sinusoids.

19) L69: "99": why exactly 99?

20) L70-71: I find the sentence a bit clumsy ("...after...after..."). Please rephrase.

21) L73: "...the injected mass..."–>"...the estimated the injected mass..."

22) L74: The nlm() function in R might be not known by many people. Please explain how the errors are calculated.

23) L80-81: Please mention these estimates. This might be given as a table and briefly discussed, like a "sensitivity analysis".

24) L85: Please mention year/month for the "Black Saturday event". Please state clearly how much water vapour was released during this event, so that it is comparable with the present study.

25) Fig. 9: It would be informative to also show the MLS observations locations (starting points of back-trajectories).

26) L153: "They attribute these anomalies to rapid ascent of tropospheric air": This is also discussed by Khaykin et al., 2020 (https://www.nature.com/articles/s43247-020-00022-5), which don't exclude that part of the ozone depletion in the fire plumes are linked to chemical processes, in addition to dynamical/vertical transport processes.

From my perspective, I don't see how a chemical component can be excluded, e.g. due to the heterogeneous chemistry linked to high aerosol content in the plume, or a radiative component, due to the strong absorption of solar radiation in the plume. Please add the reference and mention this aspect in the text.

27) Fig. 11: What are the values mentioned for MLS (0.15, 0.3, 0.8)?

28) All section 4.2 is very unclear to me. The assumptions, methodology and results are all very confusing. Assumptions are generally not justified and looks like quite arbitrary (as also mentioned by the Authors); the methodology is discussed in 1 line and not sufficient at all to understand and replicate the experiment; results (fig. 12) is very unclear or maybe it is just not well explained in the caption and text (where are "the small points"? what about the "colour according to time", e.g. which time?).

29) L175 and the "cuboid": This is very vague and imprecise. Please show the cuboid on a map and discuss of the reasons behind your choice(s) and inherent limitations/uncertainties in the results shown in Fig. 11.

30) L181-182: "Some experimentation...in time": This sentence is really really obscure. Please develop/clarify what you mean.

31) L182-185: All choices look here very arbitrary or otherwise not discussed with the sufficient care and details. Please clarify the whole thing.

32) L199-200: "Significant injections...cold front": Please justify this statement or add a reference.

33) L206-207: Please cite Khaykin et al., 2020 (see comment 26). Please discuss this work also in terms of the ascension of black carbon polluted airmasses due to fires emissions (L238-247), which is widely discussed in that paper.

34) L216-218: ''Measurements of aerosols...over Europe": Please add to the discussion the aerosol observations of Kloss et al. 2019 (see comment 9)

35) L246-247: "Once it did reach...PNE": Please justify this sentence

36) Conclusion: same comment as for the Introduction. This has to be significantly widened/rewritten.

37) L256-257: "Whether they will become...be seen": Sentences like this one are so generic that contain basically no information content. Please avoid throughout the text (i.e. be more specific, refer to previous literature, extend, etc).

38) Authors contributions: Who did the work (this is not mentioned in this section)? In addition, please check if only "suggesting many improvements to the text" is sufficient to grant a co-authorship in a scientific paper. This has to be checked with the ACP Editor as it is generally considered a very important ethical issue.

---

## Author Comment (AC1) · 30 Apr 2021

**Interactive comment* on "Stratospheric pollution from Canadian forest fires" by Hugh C. Pumphrey et al.**

Hugh C. Pumphrey et al.

April 30, 2021

Note: We use *italics* for direct quotes from the reviewers.

**Reviewer 1**

- The reviewer asks: have we tried using MAD instead of the procedure described in the paper. We have not tried this. Experiments with artificial datasets consisting of a large number of points with a Gaussian distribution, together with a small number of outliers suggests that if the outliers are fewer than 5–10% of the total points, MAD and the procedure we used give very similar results. It is possible to generate cases which fool one or the other of the approaches, but these cases tend to be those where there are more than 5–10% of outliers, heading towards the case of a bimodal distribution. And, as the reviewer says, once you are in that situation, any approach designed to detect small numbers of outliers is likely to fail. We do not propose to apply MAD in this paper.

- Page 3, Fig 2: *Are the data used in Figure 2 poleward of 25N or 15N?* The caption says both, so the reviewer is right: this is clearly a typo. The figure actually shows 15°N to 83°N. As the event reaches no further south than about 30°N we have re-made the figure to cover poleward of 27.5°N. The resulting figure looks almost exactly the same as the original.

- *I assume the oscillatory appearance of enhanced CO in Figure 3 over North America is a byproduct of the MLS sampling (and not any transport behavior), but that it does reveal the meridional extent of the plume. If true, it might be worth mentioning in the figure caption as a clarification.* The data in Fig. 3 lie along the MLS orbit tracks; more examples of the tracks can be seen in Fig. 1. We will add wording to the caption to point this out to the reader.

- *What is the semi-transparent red circle at roughly 70N and 125W in Figure 3?* We can see no such feature in Fig. 3. The point 125°W, 70°N is on the north coast of Canada; all of the data points there are blue. The only vaguely circular thing visible in that area that is not blue is the zero of "70", which is gray. Any red circle on the figure would be a point with high CO, at 215 hPa, and from a day towards the later end of the period shown.

[Figure]

Figure 1: Time-longitude plot as Fig. 4 in the discussion paper, but with the points coloured by the branch to which we have assigned them. We number the branches 1–5 for convenience. Although branches 3 and 4 only become clearly separate at day 16 we assign points to them from day 10 onwards. Points which were not assigned to a branch are plotted in gray.

- *Pg. 04, Ln. 059: "...the polluted airmass is divided into two parts on three occasions: at approximately 9 days, 16 days and 41 days after 12 August 2017" What about the split at 30 days?* The most likely explanation is that, at that point, the two parcels which split off at 16 days are crossing over each other, at the same longitude, but different altitudes and latitudes. The parcel that turns round to head westwards is between 46 and 31 hPa, and is equatorward of 38°N. The one that continues eastwards is between 68 and 46 hPa, and is poleward of 42°N . We will add text explaining this and will consider if there is a possible figure we might show that would clarify what is happening. To generate a potential figure we first identify the tracks taken by the various parts or branches of the event. This is done (Fig. 1) by identifying sufficient (time, longitude) pairs by hand to define the various branches. The branches thus defined are shown as solid lines in Fig. 1. Next, we label the data locations with enhanced CO by which branch is closest in longitude. A point is accepted as belonging to a branch if it is less than 20 degrees of longitude from the track, but only as long as it is more than 35 degrees of longitude from the next nearest track. This leaves any points that are close to two different tracks unassigned. Then we calculate the average height of the points with enhanced CO for each branch, and for all points belonging to any branch. These averages are calculated for every day, using overlapping windows which are 3.2 days wide. The result is shown in Fig. 2. After day 15, branch 4 is clearly at a higher altitude than branch 3. The exception is the few days around day 28 where branches 3 and 4 cross over; here it is not possible to reliably assign the data to a branch.

- *I was wondering what the causes of the different splits could be and whether they might play a role in the correlation between CO and H2O. For example, the split*

[Figure]

Figure 2: Altitude of points with enhanced CO as a function of time. The dots are the original data, the lines are the mean height, averaged over a 3.2 day window. Both lines and dots are coloured according to the branches identified in Fig. 1. The black line is the mean height of all points identified as being in a branch. The dashed vertical line marks the date (2017-09-09) where branches 3 and 4 cross over.

*around 9 days occurs over the Atlantic and appears to be associated with some lifting. Hurricane / Tropical Storm Gert was active around that time and location, is it possible that it played a role in this split and ascent? Similarly, the split around 30 days occurs at the border of the Tibetan plateau during the Asian Monsoon (the eastward path goes around the northern side while the westward path goes south above the plateau towards lower latitudes and is also associated with some lifting).* The split at day 9 occurs at 2017-08-21, at around 25°W, 50°N. TS Gert was south of Newfoundland, at 55°W, 40°N, at 2017-08-17 06:00. By 2017-08-18 12:00 it was at 35°W, 50°N and was merging into a larger low-pressure area. This low-pressure area was slow-moving; it was still in the area of the split at the time the split occurred, but was weak, and becoming weaker, by that time. As noted above, the event at 28–30 days is not, in our opinion, a split, but is where the paths of two parts of the polluted airmass are at the same longitude, but at different altitudes.

- *The split at 41 days is interesting because it is difficult to tell if it really is a split. Trying to follow along between Figs. 3 and 4, the paths are disjointed. It does appear that the airmass moving East over northern Japan becomes the so-called "slower part" in Fig. 4. However, the "faster part" appears about 15 degrees North of this airmass and then subsequently moves over Alaska, Northern Canada, Quebec, before reappearing over Europe. Could the disconnect be an observation gap in MLS observations or the technique for determining "enhancement" not being sensitive enough at that time? Could the split have anything to do with Typhoon Talim that was in that area at that time (and also possibly affecting the amount of $H_2O$)? Could it possibly not have been a split and instead been new material injected at that time, which might explain the small bump/deviation in Fig. 5b around late September? Does a plot similar to Fig. 4, but using latitude instead of longitude, help visualize this at all?* It is very difficult to tell where this split occurs; this is, as the reviewer

suggests, a consequence of the limited sensitivity of the measurements and their limited resolution horizontally and in time. Branch 5 (as numbered in Figs. 1 and 2) seems to appear over the date line on day 40 without any clear link to branch 3. The reviewer is correct that Typhoon Talim (by then downgraded to a tropical storm) crossed the path of branch 3 sometime around day 40. We cannot comment as to whether the storm was likely to have caused the split between branches 3 and 5. The data appear to us to be too sparse to inform the reviewer's hypothesis that the storm caused a further injection of material into the stratosphere. I have made time-latitude plots and maps, coloured by branch, but none of them clearly show where branch 5 came from.

- *Figure 5 says it shows all data poleward of 27.5N. Is there a reason this is different from Fig. 2?* As noted above, we have altered Fig. 2 to cover the same latitude range as Fig. 5.

- *Pg. 05, Ln. 068: "The seasonal behaviour of CO is fitted with a mean value and annual and semiannual sinusoids as shown in Fig. 5(a); this fit ignores the data for a period of 99 days after the PNE" Over what total date range is this fit to the seasonal cycle determined? Pg. 05, Ln. 071: ". . . we fit the decay with an exponential . . ." While often popular, the decay is almost never linear in log space and allowing for some degree of curvature would be more precise. Unfortunately, this makes a simple calculation of the decay constant impossible and likely contributes to the wide range of possibilities.* The seasonal cycle is fitted using the period shown in the figure, minus the 89 days after the fire. (The paper as discussed says 99; this is an error which we will correct.) It turns out not to help if you try using a longer period because the "seasonal" behaviour is not really seasonal at all. The reviewer is correct that the decay is probably not an exact exponential. Nevertheless, it seems the least bad way to get an estimate of the total injected mass by using the MLS data alone. We will modify the text to acknowledge the uncertainty in the mass calculation due to our choice. We also intend to modify Fig. 5, using colour to show which points are used in the seasonal cycle fit and which in the exponential fit.

- *Pg. 05, Ln. 075: ". . . presumably because the plume has a small horizontal extent and lies in between the MLS orbit tracks." Or was material was still being injected / lifted?* A good point. Given the rapid rise of the plume noted in the paper, it is possible that it takes some of those first few days to reach an altitude where MLS can see all of it. We will add wording to clarify this. Nevertheless, we think the small size of the plume combined with the large gap between the MLS orbits is the more important effect.

- *Pg. 06, Ln. 088: "These points are clearly not connected with the PNE so we have removed them from Fig. 6." What were the criteria for separation and what are the origins of these removed data? Also, the authors state that these are removed from Fig. 6, but it definitely still appears to contain data that likely have nothing to do with the polluted airmass.* As already stated in the paper, the points were removed if they were in the longitude range 60°E to 180°E and if they were at pressures greater than 130 hPa (effectively, removing the 215, 177 and 146 hPa levels). Figure 3 in this reply shows what Fig. 6 in the paper would look like without these points being removed. As the reviewer notes, there are still a few points not connected to the

[Figure]

Figure 3: Figure 6 of the paper, but without the removal of points between 60°E to 180°E and at the 215, 177 and 146 hPa levels.

polluted airmass, but they are few compared to the large number removed when constructing Fig. 6 in the paper. Unlike the PNE, the removed points are an annual feature: Fig. 4 is identical to Fig. 3 except that it is constructed for the same period of 2016. In both years most of the points with unusually high water vapour lie equatorwards of 40°N and over India, China, the Bay of Bengal and the western Pacific. They are presumably caused by the strong convection which occurs in this region in summer as an aspect of the South Asian Monsoon (see Jiang et al. (2010) (https://doi.org/10.1029/2009JD013256). We do not consider that Figs. 3 and 4 of this reply need to be included in the paper; we provide them here to support this reply to the referees.

- *Pg. 07, Ln. 093: "The algorithm used fails to detect the enhancement at this time because of the very large background variability at lower latitudes." Would a different detection method such as the median and median absolute deviation help with this?* We have not tried this but we suspect not. If you include all of the latitudes which the plume eventually visits, the $H_2O$ values within the plume are not remarkable, as we show in Fig. 5 of this reply. The only way to have any formula that detects extreme values detect the plume would be to consider only latitudes poleward of (say) 55°N for the first few days of the event.

[Figure]

Figure 4: As Fig. 3, but for 2016. The PNE is not present, but the points we filter out in order to construct Fig. 6 of the paper are present in 2016 as well.

[Figure]

Figure 5: As Fig. 1 in the discussion paper, but for $H_2O$. The event is clearly visible, but is only an unusual value if you consider latitudes poleward of $\sim 55°N$. Latitudes between 27.5°N and 55°N have many points whose $H_2O$ mixing ratios are similar to those in the polluted airmass caused by the PNE.

[Figure]

Figure 6: As Fig. 2 in the discussion paper, but for $H_2O$. Note how the event does not show up in water vapour until some days after the event and only at 178 hPa and higher altitudes.

- *Pg. 07, Ln. 095: "After about 50 days the water vapour in the polluted airmass exceeds the background value to a greater extent than is the case for CO. After 63 days, the polluted airmass can only be identified in the water vapour data; this is presumably due to the short chemical lifetime of CO in the stratosphere." Might a figure showing a time-series of $H_2O$ like Fig. 5 be useful here?* A figure like Fig. 5 in the paper turns out to be of no use for water vapour. Clear though the event is when plotted as in Fig. 6, it does not show up at all in a version of Fig. 5 made with the $H_2O$ data. Perhaps more helpful is a $H_2O$ version of Fig. 2 in the discussion paper. We show this in Fig 6 of this reply; we do not consider that the paper needs this figure.

- *[Page 10, fig 8] If HCl is an product that is considered enhanced as an artifact, then why is it included in Figure 8 (and the only one of those products included)?* On reflection, we think that the reviewer is correct and will remove HCl from Fig. 8.

- *The authors state that the computed slopes of correlation are within the expected ranges. Can the authors show the expected ranges either in Table 2 or Figure 8?* The reviewer does not give a line number for this comment. As far as we can see, everywhere that we have made this sort of remark, we have been comparing our values to the published values in the right-hand column of Table 2, some of which are plotted, with their estimated errors, in Fig. 8 of the discussion paper.

- *Pg. 12, Ln. 161: "This suggests that over these first few days the polluted airmass is closer to 215 hPa than to 147 hPa." Can the vertical resolution of MLS be playing a role here? Also, this might suggest that the airmass was rising in a way that the model was not properly accounting for (e.g., self-lofting of the black carbon in the plume).* The first point was essentially the point that we were making: the polluted airmass may well have a rather small vertical extent; too small for MLS to resolve in any detail. As long as some of the excess CO is between 215 hPa and 147 hPa it will show up in the MLS data at both of those levels. We will extend our wording at this point in the paper to make it more explicit. The second point is made later in the paper when discussing FLEXPART; the reviewer is correct that it should be made here as well.

- *Pg. 13, Ln. 190: "The injected mass is $620 \pm 80\,Gg$, considerably smaller than the value of $2400 \pm 300\,Gg$ that we obtained by the simple method discussed in section 3.1. This is qualitatively reasonable, given that the MCMC approach only considers CO that is observable by MLS near the start of the event, while the simple method of section 3.1 includes CO that was injected at too low an altitude for MLS to observe it and subsequently ascended to observable altitudes." I would assume that some of this discrepancy is also likely because the initial MLS measurements do not cover the full horizontal extent of the spread of the polluted airmass at this time. I believe this is what the authors are trying to say but it could be made more explicit. . . .* That is not the point that we were trying to make: we shall attempt to make the wording in the final paper more explicit. The point of the method in sec 3.1 is that by looking at the whole event, including days where the MLS measurements miss the polluted airmass, days where they go right through it, and days when it has spread out, we can obtain a reasonable estimate of the injected mass despite the limited horizontal resolution. The FLEXPART approach should also allow a good estimate despite the limited horizontal resolution; it is limited by the fact that FLEXPART cannot model the ascent of the polluted airmass. We will clarify these points in the revised text.

- *. . . Additionally, regarding the comment of altitudes of injection, do the MLS observations not reach the tropopause?* The lower limit of the MLS CO measurements is governed by the presence of clouds and water vapour. At the latitudes of interest here, the lowest usable level is $215\,hPa$ or about $10.5\,km$. This is close enough to the injection height we get from fitting the data to the FLEXPART results that it is probable that at the very start of the event, MLS is observing the top of the polluted airmass but not the bottom. We will make the text clearer on this point.

- *Pg. 14, Ln. 207: "The mass of CO injected into the lower stratosphere by the Black Saturday fire was, when estimated by the technique of section 3.1, 1.3 Tg – just over half of the mass injected by the PNE." Is this result from an analysis first performed in this work or does it derive from another source? If the latter, it should be referenced. If the former, then more elaboration on the analysis needs to be included than simply the result.* The Black Saturday event is described by Pumphrey et al. (2009) as referenced in the original paper, but that paper does not include an estimate of the injected mass. We will extend the wording to make it clear that the estimate we give here has not been presented before.

- *Pg. 14, Ln. 209: "The 2019/20 ANY fire injected a mass of CO considerably larger even than the PNE: initial esimates [there's a typo here], again using the method of section 3.1, place this mass in the range 8–10 Tg." In the same line as my previous comment, any additional analysis needs to elaborate more on at least the temporal and spatial extent of the data used. Also, while I understand the desire to compare the PNE to the more recent and even larger ANY event, the authors already claim to have a paper in preparation so I find it quite unusual to throw in preliminary results of that analysis in this paper.* The estimation of the total mass for the ANY event has not been presented elsewhere, including in the paper by Schwartz et al. That paper is now published, but was in preparation when this paper was submitted, and its contents were not finalised. We will add a reference to Schwartz et al. As with the estimate for the Black Saturday event, we will clarify

that the result is presented for the first time, and will explain that the estimate was made in the way described in sec 3.1 of the discussion paper. We will also replace the colon with a semicolon and correct the spelling of "estimates". We note that mass estimates for the ANY are also calculated by Khaykin et al. (2000) (`https://doi.org/10.1038/s43247-020-00022-5`) and will reference that paper.

- *Pg. 15, Ln. 232: "The location of these dots suggests that the polluted airmass observed by MLS is probably associated with Anvils 2, 3 and 4 and is less likely to be associated with Anvil 5. This in turn suggests that the emissions come from the plateau complex of fires (west of Nazko) and not from the Elephant Hill fires (north of Ashcroft)." The analysis performed here is with MLS data that is highly localized. The analysis supports the stated origin of the polluted airmass observed by MLS, but it does not rule out the other locations as potential sources that injected polluted air into regions not immediately observed by MLS.* It is, of course, always possible that the pollutants were injected by one or more pyroCb events anywhere along a trajectory between the Plateau complex of fires and the first MLS observation of enhanced CO. It is also possible that CO was injected but was at locations not observed by MLS during the days from which we used data for the MCMC analysis. We will add a sentence to point out these limitations of the analysis.

- *The entire discussion about comparing black carbon and nuclear war on page 15 (lines 240 to 247) seems very out of place in this paper.* The first-named author disagreed with some of the co-authors as to whether to include this paragraph. The first-named author remains of the opinion that the link between the event we describe and the nuclear winter hypothesis is interesting enough to include.

- *Pg. 15, Ln. 251: "The polluted airmass was much wetter than the surrounding air in the stratosphere . . ." How much wetter? There do not appear to be any quantitative results of H2O in this paper. Why does the paper discuss other biomass burning products (i.e., HCN, $CH_3CN$, $CH_3OH$, and $CH_3Cl$) and perform correlation analyses and yet only computes an injection mass for CO? This should be performed or the title changed to something centered on CO. It would be good to include some more results (both qualitative and quantitative) in the conclusion* It is clear that the reader might be interested in the total mass of other species injected by the PNE. The main reason for not estimating injected masses for species other than CO is that the method described in the paper does not work well for other species. We have already noted that this is the case for water; we have attempted to obtain injected masses of other species in the same way and have not succeeded. While such calculations are a potentially fruitful area of study, they would require too much further research for them to be included in the present paper. We note that Khaykin et al. (2000) (`https://doi.org/10.1038/s43247-020-00022-5`) provide mass estimates for some other species for the ANY event — this is possible because the injected masses for that event were considerably larger than for the PNE.

**Reviewer 2**

- *1)The originality of the manuscript must be clarified and openly discussed, in particular with respect to the previous work of Yu et al., 2019. What's new and what's*

*complementary with respect to Yu et al., 2019? This is not clear at all in the present manuscript version, at least to me.* The two papers are complementary: this paper describes the MLS observations of the event in detail while Yu et al. (2019) is a modelling paper which makes brief mention of the MLS $H_2O$ and $O_3$ data. Given that our paper references Yu et al. (2019) and notes which species are described in that paper it did not seem necessary to us to go into more detail.

- *2) The paper is unfortunately very confusing and certainly not very well crafted. The Abstract, Introduction and Conclusions are very short, incomplete, probably rushed. Many statements are not justified but just "written down". Many discussion and argumentations are just lacking. All the analyses with trajectories (Sect. 4.2) are very obscure and must be extensively clarified/rewritten. [...] The Authors should put a much larger effort in the writing of the text, the production of intelligible figures and the discussion of their results in the context of previous literature.* We accept that the abstract, introduction and conclusions should be somewhat more detailed and will expand them.

- *Many crucial references are missing.* The reviewer's general comment here provides insufficient guidance on any deficiencies in our manuscript regarding missed references. The reviewer directs us to a few papers in their detailed comments; we will, of course, include references to those papers in the revised version of our paper.

- *1) Title: The title looks like a bit too general. "Pollution" with respect to what? With which tools? Please be more specific.* What we meant was "with respect to the background levels of a variety of biomass-burning products". Any title is a tradeoff between brevity and explicitness. We agree that, with the proliferation of papers describing this event, we should change the title of this paper to something more explicit. We propose "MLS observations of biomass-burning products in the stratosphere from Canadian forest fires in August 2017"

- *2) Abstract: I feel that the Abstract is totally inadequate to represent concisely the overall scopes of the study and the results. Please rewrite it in a more extended way so to include scopes and results of the manuscript.* On reflection, we think that the reviewer is right to say that the abstract is too short. The abstract in the final paper will be substantially and appropriately enhanced.

- *3) Introduction: The section called "Introduction" is not nearly sufficient to introduce the present work. This must be significantly widened to frame the present study in its general context. Examples (not exhaustive): Why we study fires? What are their impacts on the atmosphere (perturbations with respect to background conditions)? How we study fires? What we know about British Columbia fires so far? Please extend significantly the Introduction so that this section is a real "Introduction" to the present work.* The amount of introductory material required in a scientific paper is to some extent a matter of opinion and style. We will bolster the introduction with a fuller, albeit appropriate, survey of the literature.

- *4) L10-11: "On 12August. . .(215 hPa)": Please add a reference.* The relevant reference (Peterson et al., 2018) occurs in the next sentence; we did not think it necessary to have two callouts to the same paper in consecutive sentences. We will re-word the text to make it clear that the callout applies to both sentences.

- *5) L11: "The fires are described in some detail. . .": What do you mean with "some" detail?* We will change that to "detail".

- *6) L12-14: "The polluted airmass. . .(26 hPa)": All this is described by Peterson et al.? Please clarify.* That sentence is referring to our own results later on: We will move it to the end of the introduction and make it clearer that we are referring to material later in the paper.

- *7) L16-17: In Khaykin et al. there is a very detailed description of the event, so please develop what you mean with "the global evolution of the plume".* As noted above, the introduction will be re-worded and extended.

- *9) Introduction: About the BC fire plume dispersion, the Authors are missing here an important reference about the larger scale dispersion of the plume: Kloss, C., Berthet, G., Sellitto, P., [et al.] : Transport of the 2017 Canadian wildfire plume to the tropics via the Asian monsoon circulation, Atmos. Chem. Phys., 19, 13547–13567, https://doi.org/10.5194/acp-19-13547-2019, 2019. Please add the reference and a discussion, in particular in terms of the larger scale dispersion of the BC fires plume.* We will add that reference and will discuss it later in the paper.

- *10) Introduction: A detailed description of: a. the results of Yu et al., 2019; b. what is new in the present work, with respect to Yu et al., and worth publication is required in the Introduction.* We do not agree that we need to repeat the results of Yu et al., 2019 in detail. We already state that our paper reports the full suite of biomass-burning products observed by MLS and contrast that with Yu et al., who report very briefly on only the $O_3$ and $H_2O$ data. We will clarify that our analysis goes well beyond that of Yu et al. in interpreting the MLS measurements.

- *11) Sect. 3: A general section title "Results" (joining "Observations", "Trajectories" etc) seems necessary here* We will make the current sections 4 and 5 into subsections of an overarching results section.

- *12) L34: "recommended": Recommended by whom?* By the MLS team, in the data quality document. We already reference this document; we will add a reference callout to it at this point in the text.

- *13) L34-37: Please justify these statements.* We will do this by referencing the MLS data quality document.

- *14) L40-41: "this is about. . .value": Where are these values (zonal mean; daily maximum) taken from?* From the MLS data at times outside of the event. We will make the wording clearer.

- *15) Fig. 1: Please put units next to the colorbar* We will do this; it isn't strictly necessary as the units are given in the caption. We agree with the reviewer that adding units to the figure will make understanding it more immediate.

- *16) L55-56: ". . .and the other returning westwards across the Atlantic at about 33 N.": this is probably linked to the Asian Monsoon anticyclone circulation - see Kloss et al., 2019, mentioned in a previous comment, please add discussion.*

Kloss et al.'s theme was the evidence for transport of PNE smoke around the eastern edge of the Asian monsoon anticyclone (AMA) into the tropics, where it was subsequently lifted via the Brewer Dobson circulation. The feature described in our paper did not get to the eastern edge of the AMA and never reached tropical latitudes. Moreover, it was at an altitude when it reversed course that was much greater than the aerosols visualized and analyzed by Kloss et al.: $46 - 31\,\mathrm{hPa}$ ($\theta = 540 - 600\,\mathrm{K}$) rather than $\theta = 380\,\mathrm{K}$ (about $120\,\mathrm{hPa}$). Hence it bears no resemblance to the Kloss et al. morphology, and no such discussion as suggested by the reviewer is called for.

- *17) Fig. 5: "PNE" not visible in panel a. Also, please put more human-readable dates in the $x$ axis.* We have calendar month boundaries on the upper $x$ axis AND fractional years on the lower $x$ axis. We do not understand what else we could provide in terms of human-readable dates. We will adjust the green letters PNE so that they are not obscured.

- *18) L69 and Fig. 5: Where is the annual sinusoid? I see only a semiannual one. Please be more precise in the text about periods for the fitting sinusoids.* The components used are exactly annual and semi-annual. However, the amplitude of the semiannual component is larger in this particular example. We will alter the text to make this clear.

- *19) L69: "99": why exactly 99?* We experimented with the end date and chose one which seemed to be after the event was over (as far as the total mass of CO went) and for which the seasonal cycle fit to data outside of the event appeared most convincing. Only once we had done this did we notice that the number of days was 99. But in fact a closer check shows that the value of 99 was recorded wrongly; it should have been 89. We intend to revise Fig. 5, using colour to show which days are used for the seasonal cycle fit and which days are used for the exponential fit.

- *20) L70-71: I find the sentence a bit clumsy ("...after...after..."). Please rephrase.* We will replace "after the fire" with "following the fire".

- *21) L73: ". . .the injected mass. . ."$\rightarrow$". . .the estimated [the (sic)] injected mass. . ."* We will insert "estimated" for both the mass and the decay time.

- *22) L74: The nlm() function in R might be not known by many people. Please explain how the errors are calculated.* It is acceptable to cite open-source software these days. However, the actual function used was `nls()`, not `nlm()` — we will correct this in the final version. The `nls()` function determines the least-squares estimates of the parameters of a nonlinear model. By default (as we use it) it uses the Gauss-Newton method. The errors are calculated using linear theory; this is an approximation which is useful unless the errors are very large or the problem is very nonlinear in the vicinity of the least-squares estimate.

- *23) L80-81: Please mention these estimates. This might be given as a table and briefly discussed, like a "sensitivity analysis".* We will do this in as brief a manner as seems appropriate.

- *24) L85: Please mention year/month for the "Black Saturday event". Please state clearly how much water vapour was released during this event, so that it is comparable with the present study.* We do give the date of the Black Saturday (BS) event

later on. We will add it here as this is the first mention of the event in the paper. The values of excess $H_2O$ in the BS event were small enough that they were not easily distinguishable from the background values. We will add a sentence to this effect.

- *25) Fig. 9: It would be informative to also show the MLS observations locations (starting points of back-trajectories).* Those points are on the figure already and are shown in the legend. We will re-write the caption to make it more obvious that this is the case.

- *26) L153: "They attribute these anomalies to rapid ascent of tropospheric air": This is also discussed by Khaykin et al., 2020 `https://www.nature.com/articles/s43247-020-00022-5`, which don't exclude that part of the ozone depletion in the fire plumes are linked to chemical processes, in addition to dynamical/vertical transport processes. From my perspective, I don't see how a chemical component can be excluded, e.g. due to the heterogeneous chemistry linked to high aerosol content in the plume, or a radiative component, due to the strong absorption of solar radiation in the plume. Please add the reference and mention this aspect in the text.* Khaykin et al., 2020 (on which the reviewer is a co-author) came out at about the time that we were finalising the paper under review. We will add the reference; we apologise to him for missing it out on the discussion paper.

- *27) Fig. 11: What are the values mentioned for MLS (0.15, 0.3, 0.8)?* As the caption explains, these are also CO mixing ratios in ppmv. We will either expand the caption to make it more explicit or change the legend entries to read (e.g.) MLS CO < 0.15 ppmv.

- *28) All section 4.2 [the FLEXPART section] is very unclear to me. The assumptions, methodology and results are all very confusing. Assumptions are generally not justified and looks like quite arbitrary (as also mentioned by the Authors); the methodology is discussed in 1 line and not sufficient at all to understand and replicate the experiment [...]* We will expand the description of the approach to the extent necessary to ensure clarity and reproducibility.

- *[...] results (fig. 12) is very unclear or maybe it is just not well explained in the caption and text (where are "the small points"? what about the "colour according to time", e.g. which time?).* This is explained in the text, as the caption says, but not until the discussion section. We will either extend the caption to duplicate this information or signpost the discussion better.

- *29) L175 and the "cuboid": This is very vague and imprecise. Please show the cuboid on a map and discuss of the reasons behind your choice(s) and inherent limitations/uncertainties in the results shown in Fig. 11.* Much of this is discussed in the next paragraph in the manuscript. It would be straightforward to add a rectangle to show the horizontal extent of the cuboid on Fig. 11 or Fig. 12; we will try this and include it in the final paper if it seems to improve clarity.

- *30) L181-182: "Some experimentation. . .in time": This sentence is really really obscure. Please develop/clarify what you mean.* What we mean is that if one attempts to estimate all of the nine properties (mass, plus location in 4 dimensions

plus size in four dimensions) which define the release, then the inverse method fails. By "fails" we mean that it produces solutions which are very widely spread in parameter space, many of which do not make sense in view of what we know about the event from other types of data. If, instead, we attempt to estimate only six parameters, fixing the horizontal size and duration in time of the release, then the inverse method produces sensible results for those six parameters. We will reword this part of the description so that the process is described in more detail.

- *31) L182-185: All choices look here very arbitrary or otherwise not discussed with the sufficient care and details. Please clarify the whole thing.* These sentences should become clearer with the more detailed description of the approach as outlined in the response to the previous comment.

- *32) L199-200: "Significant injections. . .cold front": Please justify this statement or add a reference.* As earlier in the paper, Peterson et al. (2018) is the obvious reference and occurs in the following sentence. As before, we will re-word the text so that it is clearer that the Peterson et al. (2018) reference applies to both sentences.

- *33) L206-207: Please cite Khaykin et al., 2020 (see comment 26). Please discuss this work also in terms of the ascension of black carbon polluted airmasses due to fires emissions (L238-247), which is widely discussed in that paper.* We note above that we will add references to Khaykin et al., 2020 where appropriate.

- *34) L216-218: "Measurements of aerosols. . .over Europe": Please add to the discussion the aerosol observations of Kloss et al. 2019 (see comment 9)* We will reference Kloss et al. at this point in the paper.

- *35) L246-247: "Once it did reach. . .PNE": Please justify this sentence* The sentence is just a corollary of the preceding sentence, so it is justified by the two references in the sentence before that. There is a balance to be struck between inadequate referencing and the cluttering up of the text with an excess of repeated reference callouts. It is clear that we and the reviewer feel that balance to lie in slightly different places.

- *36) Conclusion: same comment as for the Introduction. This has to be significantly widened/rewritten.* We will expand the conclusions section as noted earlier in this reply.

- *37) L256-257: "Whether they will become. . .be seen": Sentences like this one are so generic that contain basically no information content. Please avoid throughout the text (i.e. be more specific, refer to previous literature, extend, etc).* The reviewer is correct in a sense that the sentence carries no information. In another sense the reviewer is wrong: it appears to me to be a valid point that we can not yet state that events such as the one described are becoming commoner, but that we should be able to make such a statement in the future, once we have more data. What is more, it is important to make this statement to prevent readers assuming that events of the type described provide a clear signal of climate change.

- *38) Authors contributions: Who did the work (this is not mentioned in this section)? In addition, please check if only "suggesting many improvements to the text" is sufficient to grant a co-authorship in a scientific paper. This has to be checked with*

*the ACP Editor as it is generally considered a very important ethical issue.* We will make the contributions section more explicit in a number of ways. All of the authors have contributed in some way; we do not intend to remove any of them from the author list.

**Other changes**

- A reader has contacted us to note a couple of missing references. They were different missing references from those identified by reviewer 2. We will add these references to the final paper.

- We note that part of the coastline of Alaska is missing from Fig. 3 in the paper. We have corrected this. We have also changed the symbol used for 46 hPa from an asterisk to a square. This is to ensure that it does not resemble the symbols for 147 and 100 hPa superimposed.

---

## Author Response (AR1)

This version typeset on June 2, 2021
Note: We use *italics* for direct quotes from the reviewers.

**Reviewer 1**

- The reviewer asks: have we tried using MAD instead of the procedure described in the paper. We have not tried this. Experiments with artificial datasets consisting of a large number of points with a Gaussian distribution, together with a small number of outliers suggests that if the outliers are fewer than 5–10% of the total points, MAD and the procedure we used give very similar results. It is possible to generate cases which fool one or the other of the approaches, but these cases tend to be those where there are more than 5–10% of outliers, heading towards the case of a bimodal distribution. And, as the reviewer says, once you are in that situation, any approach designed to detect small numbers of outliers is likely to fail. We do not propose to apply MAD in this paper.

- Page 3, Fig 2: *Are the data used in Figure 2 poleward of 25N or 15N?* The caption says both, so the reviewer is right: this is clearly a typo. The figure actually showed 15°N to 83°N. As the event reaches no further south than about 30°N we have re-made the figure to cover poleward of 27.5°N. This improves consistency with later figures. The resulting figure looks almost exactly the same as the original.

- *I assume the oscillatory appearance of enhanced CO in Figure 3 over North America is a byproduct of the MLS sampling (and not any transport behavior), but that it does reveal the meridional extent of the plume. If true, it might be worth mentioning in the figure caption as a clarification.* The data in Fig. 3 lie along the MLS orbit tracks; more examples of the tracks can be seen in Fig. 1. We have added wording to the caption to point this out to the reader.

- *What is the semi-transparent red circle at roughly 70N and 125W in Figure 3?* We can see no such feature in Fig. 3. The point 125°W, 70°N is on the north coast of Canada; all of the data points there are blue. The only vaguely circular thing visible in that area that is not blue is the zero of "70", which is grey. Any red circle on the figure would be a point with high CO, at 215 hPa, and from a day towards the later end of the period shown.

- *Pg. 04, Ln. 059: "... the polluted airmass is divided into two parts on three occasions: at approximately 9 days, 16 days and 41 days after 12 August 2017"* What about the split at 30 days?* The "split near 30 days" is not really a split: two parcels which split off at 16 days are crossing over each other on the figure. They are at the same longitude, but different altitudes and latitudes. The parcel that turns round to head westwards is between 46 and 31 hPa, and is equatorward of 38°N. The one that continues eastwards is between 68 and 46 hPa, and is poleward of 42°N. We have added text explaining this and have considered if there is a possible figure we might show that would clarify what is happening. To generate a potential figure we first identify the tracks taken by the various parts or branches of the event. This is done (Fig. 1) by identifying sufficient (time, longitude) pairs by hand to define the various branches. The branches thus defined are shown as solid lines in Fig. 1. Next, we label the data locations with enhanced CO by which branch is closest in

[Figure]

Figure 1: Time-longitude plot as Fig. 4 in the discussion paper, but with the points coloured by the branch to which we have assigned them. We label the branches using the nomenclature of Lestrelin et al. (2021), adding an extra branch (X). Although branches A and B1 only become clearly separate at day 16 we assign points to them from day 10 onward. Points which were not assigned to a branch are plotted in grey.

longitude. A point is accepted as belonging to a branch if it is less than 20 degrees of longitude from the track, but only as long as it is more than 35 degrees of longitude from the next nearest track. This leaves any points that are close to two different tracks unassigned. Then we calculate the average height of the points with enhanced CO for each branch, and for all points belonging to any branch. These averages are calculated for every day, using overlapping windows which are 3.2 days wide. The result is shown in Fig. 2. After day 15, branch A is clearly at a higher altitude than branch B1. The exception is the few days around day 28 where branches A and B1 cross over; here it is not possible to reliably assign the data to a branch using time and longitude alone. We have elected not to include these extra figures in the paper. We have, however, labelled the branches in Fig.4 to enable the text to be made clearer. All branches except one (branch X) have been identified by Lestrelin et al. (2021) which was published as we made the corrections to this paper. We have adapted Lestrelin's nomenclature for consistency.

- *I was wondering what the causes of the different splits could be and whether they might play a role in the correlation between CO and H2O. For example, the split around 9 days occurs over the Atlantic and appears to be associated with some lifting. Hurricane / Tropical Storm Gert was active around that time and location, is it possible that it played a role in this split and ascent? Similarly, the split around 30 days occurs at the border of the Tibetan plateau during the Asian Monsoon (the eastward path goes around the northern side while the westward path goes south above the plateau towards lower latitudes and is also associated with some lifting). The split at day 9 occurs at 2017-08-21, at around 25°W, 50°N. TS Gert was south of Newfoundland, at 55°W, 40°N, at 2017-08-17 06:00. By 2017-08-18 12:00 it was at*

[Figure]

Figure 2: Altitude of points with enhanced CO as a function of time. The dots are the original data, the lines are the mean height, averaged over a 3.2 day window. Both lines and dots are coloured according to the branches identified in Fig. 1. The black line is the mean height of all points identified as being in a branch. The dashed vertical line marks the date (2017-09-09) where branches A and B1 appear to cross over in Fig. 1.

35°W, 50°N and was merging into a larger low-pressure area. This low-pressure area was slow-moving; it was still in the area of the split at the time the split occurred, but was weak, and becoming weaker, by that time. As noted above, the event at 28–30 days is not a split, but is where the paths of two parts of the polluted airmass are at the same longitude, but at different altitudes and latitudes. We have not added any discussion of these points to the text.

- *The split at 41 days is interesting because it is difficult to tell if it really is a split. Trying to follow along between Figs. 3 and 4, the paths are disjointed. It does appear that the airmass moving East over northern Japan becomes the so-called "slower part" in Fig. 4. However, the "faster part" appears about 15 degrees North of this airmass and then subsequently moves over Alaska, Northern Canada, Quebec, before reappearing over Europe. Could the disconnect be an observation gap in MLS observations or the technique for determining "enhancement" not being sensitive enough at that time? Could the split have anything to do with Typhoon Talim that was in that area at that time (and also possibly affecting the amount of $H_2O$)? Could it possibly not have been a split and instead been new material injected at that time, which might explain the small bump/deviation in Fig. 5b around late September? Does a plot similar to Fig. 4, but using latitude instead of longitude, help visualize this at all?* It is very difficult to tell where this split occurs; this is, as the reviewer suggests, a consequence of the limited sensitivity of the measurements and their limited resolution horizontally and in time. Branch B2 (as numbered in Figs. 1 and 2) seems to appear over the date line on day 40 without any clear link to branch B1. The reviewer is correct that Typhoon Talim (by then downgraded to a tropical storm) crossed the path of branch B1 sometime around day 40. Using the MLS data, we cannot comment as to whether the storm was likely to have caused the split between branches B1 and B2. The data appear to us to be too sparse to inform

the reviewer's hypothesis that the storm caused a further injection of material into the stratosphere. I have made time-latitude plots and maps, coloured by branch, but none of them clearly show where branch B2 came from. However, Lestrelin et al. (2021) show that B2 and B1 split apart at an earlier time, and travel at similar longitudes until a time around day 40. This implies that Typhoon Talim is not responsible.

- *Figure 5 says it shows all data poleward of 27.5N. Is there a reason this is different from Fig. 2?* As noted above, we have altered Fig. 2 to cover the same latitude range as Fig. 5.

- *Pg. 05, Ln. 068: "The seasonal behaviour of CO is fitted with a mean value and annual and semiannual sinusoids as shown in Fig. 5(a); this fit ignores the data for a period of 99 days after the PNE" Over what total date range is this fit to the seasonal cycle determined? Pg. 05, Ln. 071: ". . . we fit the decay with an exponential . . ." While often popular, the decay is almost never linear in log space and allowing for some degree of curvature would be more precise. Unfortunately, this makes a simple calculation of the decay constant impossible and likely contributes to the wide range of possibilities.* The seasonal cycle is fitted using the period shown in the figure, minus the 93 days after the fire. (The paper as discussed says 99; this is an error which we have corrected.) It turns out not to help if you try using a longer period because the "seasonal" behaviour is not really seasonal at all. The reviewer is correct that the decay is probably not an exact exponential. Nevertheless, it seems the least bad way to get an estimate of the total injected mass by using the MLS data alone. We have re-visited the calculation and have modified the text to acknowledge the uncertainty in the mass calculation due to our choice. We have modified Fig. 5, using colour to show which points are used in the seasonal cycle fit and which in the exponential fit. We have also stated explicitly that we applied the calculation to three other events; we now show the results in a table.

- *Pg. 05, Ln. 075: ". . . presumably because the plume has a small horizontal extent and lies in between the MLS orbit tracks." Or was material was still being injected / lifted?* A good point. Given the rapid rise of the plume noted in the paper, it is possible that it takes some of those first few days to reach an altitude where MLS can see all of it. We have added wording to include this possibility. Nevertheless, we think the small size of the plume combined with the large gap between the MLS orbits is the more important effect.

- *Pg. 06, Ln. 088: "These points are clearly not connected with the PNE so we have removed them from Fig. 6." What were the criteria for separation and what are the origins of these removed data? Also, the authors state that these are removed from Fig. 6, but it definitely still appears to contain data that likely have nothing to do with the polluted airmass.* As already stated in the paper, the points were removed if they were in the longitude range 60°E to 180°E and if they were at pressures greater than 130 hPa (effectively, removing the 215, 177 and 146 hPa levels). Figure 3 in this reply shows what Fig. 6 in the paper would look like without these points being removed. As the reviewer notes, there are still a few points not connected to the polluted airmass, but they are few compared to the large number removed when constructing Fig. 6 in the paper. Unlike the PNE, the removed points are

[Figure]

Figure 3: Figure 6 of the paper, but without the removal of points between 60°E to 180°E and at the 215, 177 and 146 hPa levels.

an annual feature: Fig. 4 is identical to Fig. 3 except that it is constructed for the same period of 2016. In both years most of the points with unusually high water vapour lie equatorward of 40°N and over India, China, the Bay of Bengal and the western Pacific. They are presumably caused by the strong convection which occurs in this region in summer as an aspect of the Asian Summer Monsoon (see Jiang et al. (2010) (`https://doi.org/10.1029/2009JD013256`). We do not consider that Figs. 3 and 4 of this reply need to be included in the paper; we provide them here to support this reply to the referees. We have noted where the points we have removed come from and have been very explicit as to how we removed them.

- *Pg. 07, Ln. 093: "The algorithm used fails to detect the enhancement at this time because of the very large background variability at lower latitudes." Would a different detection method such as the median and median absolute deviation help with this?* We have not tried this but we suspect not. If you include all of the latitudes which the plume eventually visits, the $H_2O$ values within the plume are not remarkable, as we show in Fig. 5 of this reply. The only way to have any formula that detects extreme values detect the plume would be to consider only latitudes poleward of (say) 55°N for the first few days of the event.

- *Pg. 07, Ln. 095: "After about 50 days the water vapour in the polluted airmass*

[Figure]

Figure 4: As Fig. 3, but for 2016. The PNE is not present, but the points we filter out in order to construct Fig. 6 of the paper are present in 2016 as well.

[Figure]

Figure 5: As Fig. 1 in the discussion paper, but for $H_2O$. The event is clearly visible, but is only an unusual value if you consider latitudes poleward of $\sim 55°N$. Latitudes between 27.5°N and 55°N have many points whose $H_2O$ mixing ratios are similar to those in the polluted airmass caused by the PNE.

[Figure]

Figure 6: As Fig. 2 in the discussion paper, but for $H_2O$. Note how the event does not show up in water vapour until some days after the event and only at $178\,hPa$ and higher altitudes.

*exceeds the background value to a greater extent than is the case for CO. After 63 days, the polluted airmass can only be identified in the water vapour data; this is presumably due to the short chemical lifetime of CO in the stratosphere."* Might a figure showing a time-series of $H_2O$ like Fig. 5 be useful here? A figure like Fig. 5 in the paper turns out to be of no use for water vapour. Clear though the event is when plotted as in Fig. 6, it does not show up at all in a version of Fig. 5 made with the $H_2O$ data. Perhaps more helpful is a $H_2O$ version of Fig. 2 in the discussion paper. We show this in Fig 6 of this reply; we do not consider that the paper needs this figure.

- *[Page 10, fig 8] If HCl is an product that is considered enhanced as an artifact, then why is it included in Figure 8 (and the only one of those products included)?* On reflection, we think that the reviewer is correct; we have removed HCl from Fig. 8.

- *The authors state that the computed slopes of correlation are within the expected ranges. Can the authors show the expected ranges either in Table 2 or Figure 8?* The reviewer does not give a line number for this comment. As far as we can see, everywhere that we have made this sort of remark, we have been comparing our values to the published values in the right-hand column of Table 2, some of which are plotted, with their estimated errors, in Fig. 8 of the discussion paper.

- *Pg. 12, Ln. 161: "This suggests that over these first few days the polluted airmass is closer to 215 hPa than to 147 hPa."* Can the vertical resolution of MLS be playing a role here? Also, this might suggest that the airmass was rising in a way that the model was not properly accounting for (e.g., self-lofting of the black carbon in the plume).* The first point was essentially the point that we were making: the polluted airmass may well have a rather small vertical extent; too small for MLS to resolve in any detail. As long as some of the excess CO is between $215\,hPa$ and $147\,hPa$ it will show up in the MLS data at both of those levels. We have extended our wording at this point in the paper to make it more explicit. The second point is made later in the paper when discussing FLEXPART; we now make it here as well.

- *Pg. 13, Ln. 190: "The injected mass is $620 \pm 80\,Gg$, considerably smaller than the value of $2400 \pm 300\,Gg$ that we obtained by the simple method discussed in section*

*3.1. This is qualitatively reasonable, given that the MCMC approach only considers CO that is observable by MLS near the start of the event, while the simple method of section 3.1 includes CO that was injected at too low an altitude for MLS to observe it and subsequently ascended to observable altitudes." I would assume that some of this discrepancy is also likely because the initial MLS measurements do not cover the full horizontal extent of the spread of the polluted airmass at this time. I believe this is what the authors are trying to say but it could be made more explicit. ...* That is not the point that we were trying to make. The point of the method in sec 3.1 is that by looking at the whole event, including days where the MLS measurements miss the polluted airmass, days where they go right through it, and days when it has spread out, we can obtain a reasonable estimate of the injected mass despite the limited horizontal resolution. The FLEXPART approach should also allow a good estimate despite the limited horizontal resolution; it is limited by the fact that FLEXPART cannot model the ascent of the polluted airmass. We have attempted to clarify these points in the revised text.

- *... Additionally, regarding the comment of altitudes of injection, do the MLS observations not reach the tropopause?* The lower limit of the MLS CO measurements is governed by the presence of clouds and water vapour. At the latitudes of interest here, the lowest usable level is 215 hPa or about 10.5 km. This is close enough to the injection height we get from fitting the data to the FLEXPART results that it is probable that at the very start of the event, MLS is observing the top of the polluted airmass but not the bottom. We made the text more explicit on this point in answering one of the referees' earlier comments.

- *Pg. 14, Ln. 207: "The mass of CO injected into the lower stratosphere by the Black Saturday fire was, when estimated by the technique of section 3.1, 1.3 Tg – just over half of the mass injected by the PNE." Is this result from an analysis first performed in this work or does it derive from another source? If the latter, it should be referenced. If the former, then more elaboration on the analysis needs to be included than simply the result.* The Black Saturday event is described by Pumphrey et al. (2009) as referenced in the original paper, but Pumphrey et al. (2009) do not include an estimate of the injected mass. We have re-written the section describing the mass estimates to refer the reader to the results which we now present in an earlier section. We have done the analysis more carefully; the results presented in the discussion section are now less vague as a result.

- *Pg. 14, Ln. 209: "The 2019/20 ANY fire injected a mass of CO considerably larger even than the PNE: initial esimates [there's a typo here], again using the method of section 3.1, place this mass in the range 8–10 Tg." In the same line as my previous comment, any additional analysis needs to elaborate more on at least the temporal and spatial extent of the data used. Also, while I understand the desire to compare the PNE to the more recent and even larger ANY event, the authors already claim to have a paper in preparation so I find it quite unusual to throw in preliminary results of that analysis in this paper.* The estimation of the total mass for the ANY event has not been presented elsewhere, including in the paper by Schwartz et al. That paper is now published, but was in preparation when this paper was submitted, and its contents were not finalised. We have added references to Schwartz et al. As with the estimate for the Black Saturday event, we have clarified that the result is

presented for the first time, and direct the reader to the added material earlier in the paper showing the results. We have also corrected the spelling of "estimates".

- *Pg. 15, Ln. 232: "The location of these dots suggests that the polluted airmass observed by MLS is probably associated with Anvils 2, 3 and 4 and is less likely to be associated with Anvil 5. This in turn suggests that the emissions come from the plateau complex of fires (west of Nazko) and not from the Elephant Hill fires (north of Ashcroft)." The analysis performed here is with MLS data that is highly localized. The analysis supports the stated origin of the polluted airmass observed by MLS, but it does not rule out the other locations as potential sources that injected polluted air into regions not immediately observed by MLS.* It is, of course, always possible that the pollutants were injected by one or more pyroCb events anywhere along a trajectory between the Plateau complex of fires and the first MLS observation of enhanced CO. It is also possible that CO was injected but was at locations not observed by MLS during the days from which we used data for the MCMC analysis. We have added a sentence to point out these limitations of the analysis.

- *The entire discussion about comparing black carbon and nuclear war on page 15 (lines 240 to 247) seems very out of place in this paper.* The first-named author disagreed with some of the co-authors as to whether to include this paragraph. The first-named author remains of the opinion that the link between the event we describe and the nuclear winter hypothesis is interesting enough to include.

- *Pg. 15, Ln. 251: "The polluted airmass was much wetter than the surrounding air in the stratosphere . . ." How much wetter? There do not appear to be any quantitative results of H2O in this paper. Why does the paper discuss other biomass burning products (i.e., HCN, CH₃CN, CH₃OH, and CH₃Cl) and perform correlation analyses and yet only computes an injection mass for CO? This should be performed or the title changed to something centered on CO. It would be good to include some more results (both qualitative and quantitative) in the conclusion.* It is clear that the reader might be interested in the total mass of other species injected by the PNE. The main reason for not estimating injected masses for species other than CO is that the method described in the paper does not work well for other species. We added a note to this effect in the section describing the method. While such calculations are a potentially fruitful area of study, they would require too much further research for them to be included in the present paper. We have also added a note that Khaykin et al. (2000) (`https://doi.org/10.1038/s43247-020-00022-5`) provide mass estimates for some other species for the ANY event — this is possible because the injected masses for that event were considerably larger than for the PNE.

**Reviewer 2**

- *1)The originality of the manuscript must be clarified and openly discussed, in particular with respect to the previous work of Yu et al., 2019. What's new and what's complementary with respect to Yu et al., 2019? This is not clear at all in the present manuscript version, at least to me.* The two papers are complementary: this paper describes the MLS observations of the event in detail while Yu et al. (2019) is a modelling paper which makes brief mention of the MLS $H_2O$ and $O_3$ data. Given

that our paper references Yu et al. (2019) and notes which species are described in that paper it did not seem necessary to us to go into more detail.

- *2) The paper is unfortunately very confusing and certainly not very well crafted. The Abstract, Introduction and Conclusions are very short, incomplete, probably rushed. Many statements are not justified but just "written down". Many discussion and argumentations are just lacking. All the analyses with trajectories (Sect. 4.2) are very obscure and must be extensively clarified/rewritten. [...] The Authors should put a much larger effort in the writing of the text, the production of intelligible figures and the discussion of their results in the context of previous literature.* We have expanded and added more details to the abstract, introduction, and conclusions.

- *Many crucial references are missing.* The reviewer's general comment here provides insufficient guidance on any deficiencies in our manuscript regarding missed references. The reviewer directs us to a few papers in their detailed comments; we have included references to those papers in the revised version of our paper.

- *1) Title: The title looks like a bit too general. "Pollution" with respect to what? With which tools? Please be more specific.* What we meant was "with respect to the background levels of a variety of biomass-burning products". Any title is a trade-off between brevity and explicitness. We agree that, with the proliferation of papers describing this event, we should change the title of this paper to something more explicit. We decided on "MLS observations of biomass-burning products in the stratosphere from Canadian forest fires in August 2017."

- *2) Abstract: I feel that the Abstract is totally inadequate to represent concisely the overall scopes of the study and the results. Please rewrite it in a more extended way so to include scopes and results of the manuscript.* On reflection, we think that the reviewer is right to say that the abstract was too short. The abstract in the revised paper has been substantially and appropriately enhanced.

- *3) Introduction: The section called "Introduction" is not nearly sufficient to introduce the present work. This must be significantly widened to frame the present study in its general context. Examples (not exhaustive): Why we study fires? What are their impacts on the atmosphere (perturbations with respect to background conditions)? How we study fires? What we know about British Columbia fires so far? Please extend significantly the Introduction so that this section is a real "Introduction" to the present work.* The amount of introductory material required in a scientific paper is to some extent a matter of opinion and style. We have bolstered the introduction with a fuller, albeit appropriate, survey of the literature.

- *4) L10-11: "On 12August. . .(215 hPa)": Please add a reference.* The relevant reference (Peterson et al., 2018) occurs in the next sentence; we did not think it necessary to have two callouts to the same paper in consecutive sentences. We have changed the sentence breaks to make it clear that the callout applies from the point identified by the referee.

- *5) L11: "The fires are described in some detail. . .": What do you mean with "some" detail?* We have changed "some detail" to "detail".

- *6) L12-14: "The polluted airmass. . .(26 hPa)": All this is described by Peterson et al.? Please clarify.* That sentence was referring to our own results later on. However the content of the sentence is reported by Khaykin et al (2018). We already reference that paper but have added a callout to it at this point.

- *7) L16-17: In Khaykin et al. there is a very detailed description of the event, so please develop what you mean with "the global evolution of the plume".* As noted above, the introduction has been re-worded and extended. We do not think it necessary to repeat the description given by Khaykin (2018) at this point.

- *9) Introduction: About the BC fire plume dispersion, the Authors are missing here an important reference about the larger scale dispersion of the plume: Kloss, C., Berthet, G., Sellitto, P., [et al.] : Transport of the 2017 Canadian wildfire plume to the tropics via the Asian monsoon circulation, Atmos. Chem. Phys., 19, 13547–13567, `https://doi.org/10.5194/acp-19-13547-2019`, 2019. Please add the reference and a discussion, in particular in terms of the larger scale dispersion of the BC fires plume.* We have added that reference and discuss it later in the paper.

- *10) Introduction: A detailed description of: a. the results of Yu et al., 2019; b. what is new in the present work, with respect to Yu et al., and worth publication is required in the Introduction.* We do not agree that we need to repeat the results of Yu et al., 2019 in detail. We already state that our paper reports the full suite of biomass-burning products observed by MLS and contrast that with Yu et al., who report very briefly on only the $O_3$ and $H_2O$ data. We have clarified that our analysis goes well beyond that of Yu et al. in interpreting the MLS measurements.

- *11) Sect. 3: A general section title "Results" (joining "Observations", "Trajectories" etc) seems necessary here* We have made the current sections 4 and 5 into subsections of an overarching "Results" section.

- *12) L34: "recommended": Recommended by whom?* By the MLS team, in the data quality document. We already reference this document; we have added a reference callout to it at this point in the text.

- *13) L34-37: Please justify these statements.* We regard the reference to the MLS data quality document to be sufficient for this purpose.

- *14) L40-41: "this is about. . .value": Where are these values (zonal mean; daily maximum) taken from?* From the MLS data at times outside of the event. We have made the wording of this sentence even more explicit.

- *15) Fig. 1: Please put units next to the colorbar* We have done this; it isn't strictly necessary as the units are given in the caption. We agree with the reviewer that adding units to the figure will make understanding it more immediate.

- *16) L55-56: ". . .and the other returning westwards across the Atlantic at about 33 N.": this is probably linked to the Asian Monsoon anticyclone circulation - see Kloss et al., 2019, mentioned in a previous comment, please add discussion.*

    Kloss et al.'s theme was the evidence for transport of PNE smoke around the eastern edge of the Asian monsoon anticyclone (AMA) into the tropics, where it was subsequently lifted via the Brewer Dobson circulation. The feature described in our paper

did not get to the eastern edge of the AMA and never reached tropical latitudes. Moreover, it was at an altitude when it reversed course that was much greater than the aerosols visualised and analysed by Kloss et al.: $46 - 31\,\text{hPa}$ ($\theta = 540\text{--}600\,\text{K}$) rather than $\theta = 380\,\text{K}$ (about $120\,\text{hPa}$). Hence it bears no resemblance to the Kloss et al. morphology. We do not think a discussion is warranted at this point in the paper; we add one sentence with a reference in the discussion section.

- *17) Fig. 5: "PNE" not visible in panel a. Also, please put more human-readable dates in the x axis.* We have calendar month boundaries on the upper $x$ axis AND fractional years on the lower $x$ axis. We do not understand what else we could provide in terms of human-readable dates. We have adjusted the green letters "PNE" so that they are not obscured. We have also coloured the points to show which points are used for which stage of the analysis.

- *18) L69 and Fig. 5: Where is the annual sinusoid? I see only a semiannual one. Please be more precise in the text about periods for the fitting sinusoids.* The components used are exactly annual and semi-annual. However, the amplitude of the semiannual component is larger in this particular example. We have expanded the figure caption to explain this.

- *19) L69: "99": why exactly 99?* We experimented with the end date and chose one which seemed to be after the event was over (as far as the total mass of CO went) and for which the seasonal cycle fit to data outside of the event appeared most convincing. Only once we had done this did we notice that the number of days was 99. In order to respond to Referee 1's comments we have expanded the discussion in this area considerably and have re-visited the analysis in some detail.

- *20) L70-71: I find the sentence a bit clumsy ("...after...after..."). Please rephrase.* We have replaced "after the fire" with "following the fire".

- *21) L73: ". . .the injected mass. . ."→". . .the estimated [the (sic)] injected mass. . ."* We have inserted "estimated" for both the mass and the decay time.

- *22) L74: The nlm() function in R might be not known by many people. Please explain how the errors are calculated.* It is acceptable to cite open-source software these days. However, the actual function used was `nls()`, not `nlm()` — we have corrected this. The `nls()` function determines the least-squares estimates of the parameters of a nonlinear model. By default (as we use it) it uses the Gauss-Newton method. The errors are calculated using linear theory; this is an approximation which is useful unless the errors are very large or the problem is very nonlinear in the vicinity of the least-squares estimate.

- *23) L80-81: Please mention these estimates. This might be given as a table and briefly discussed, like a "sensitivity analysis".* In response to reviewer 1's comments we have already added a table showing the error estimates for the PNE and also for three other events.

- *24) L85: Please mention year/month for the "Black Saturday event". Please state clearly how much water vapour was released during this event, so that it is comparable with the present study.* The year and month for Black Saturday now occur earlier in the paper on account of material added to address reviewer 1's comments.

The polluted airmass from the BS event did not appear to be clearly wetter than the surrounding air; it is not possible to give a total mass of water vapour injected by the event. We have not succeeded in obtaining a good estimate for the PNE, but an estimate can be obtained for the much larger ANY event as shown by Khaykin et al. (2020).

- *25) Fig. 9: It would be informative to also show the MLS observations locations (starting points of back-trajectories).* Those points are on the figure already and are shown in the legend. We have re-written the caption to make it more obvious that this is the case.

- *26) L153: "They attribute these anomalies to rapid ascent of tropospheric air": This is also discussed by Khaykin et al., 2020 `https://www.nature.com/articles/s43247-020-00022-5`, which don't exclude that part of the ozone depletion in the fire plumes are linked to chemical processes, in addition to dynamical/vertical transport processes. From my perspective, I don't see how a chemical component can be excluded, e.g. due to the heterogeneous chemistry linked to high aerosol content in the plume, or a radiative component, due to the strong absorption of solar radiation in the plume. Please add the reference and mention this aspect in the text.* Khaykin et al. (2020) (on which we note that the reviewer is a co-author) came out at about the time that we were finalising the paper under review. We have added the reference, with a callout at this point in the text; we apologise to him for overlooking it in the discussion paper. However, I note here that Khaykin et al. (2020) do not discuss the chemical depletion in any detail at all; they merely express the opinion that it must be significant.

- *27) Fig. 11: What are the values mentioned for MLS (0.15, 0.3, 0.8)?* As the caption explains, these are also CO mixing ratios in ppmv. We have expanded the caption to make it more explicit; we did not want to either make the legend text smaller or the legend wider.

- *28) All section 4.2 [the FLEXPART section] is very unclear to me. The assumptions, methodology and results are all very confusing. Assumptions are generally not justified and looks like quite arbitrary (as also mentioned by the Authors); the methodology is discussed in 1 line and not sufficient at all to understand and replicate the experiment [...]* We have expanded the description of the approach to the extent necessary to ensure clarity and reproducibility.

- *[...] results (fig. 12) is very unclear or maybe it is just not well explained in the caption and text (where are "the small points"? what about the "colour according to time", e.g. which time?).* This is explained in the text, as the caption says, but not until the discussion section. We have added the explanation of the colours used to the caption.

- *29) L175 and the "cuboid": This is very vague and imprecise. Please show the cuboid on a map and discuss of the reasons behind your choice(s) and inherent limitations/uncertainties in the results shown in Fig. 11.* Much of this is discussed in the next paragraph in the manuscript. We have added the outline of the cuboid to Fig. 11 and noted in the text that each of the coloured dots represents the centre of such a box.

- *30) L181-182: "Some experimentation. . .in time": This sentence is really really obscure. Please develop/clarify what you mean.* What we mean is that if one attempts to estimate all of the nine properties (mass, plus location in 4 dimensions plus size in four dimensions) which define the release, then the inverse method fails. By "fails" we mean that it produces solutions which are very widely spread in parameter space, many of which do not make sense in view of what we know about the event from other types of data. If, instead, we attempt to estimate only six parameters, fixing the horizontal size and duration in time of the release, then the inverse method produces sensible results for those six parameters. We have reworded this part of the description so that the process is described in more detail.

- *31) L182-185: All choices look here very arbitrary or otherwise not discussed with the sufficient care and details. Please clarify the whole thing.* These sentences should become clearer with the more detailed description of the approach as outlined in the response to the previous comment.

- *32) L199-200: "Significant injections. . .cold front": Please justify this statement or add a reference.* As earlier in the paper, Peterson et al. (2018) is the obvious reference and occurs in the following sentence. We have inserted an extra callout for Peterson et al. (2018).

- *33) L206-207: Please cite Khaykin et al., 2020 (see comment 26). Please discuss this work also in terms of the ascension of black carbon polluted airmasses due to fires emissions (L238-247), which is widely discussed in that paper.* We have added references to Khaykin et al. (2020) in various appropriate places.

- *34) L216-218: "Measurements of aerosols. . .over Europe": Please add to the discussion the aerosol observations of Kloss et al. 2019 (see comment 9)* We have inserted a sentence referencing Kloss et al. at this point.

- *35) L246-247: "Once it did reach. . .PNE": Please justify this sentence* The sentence is just a corollary of the preceding sentence, so it is justified by the two references in the sentence before that. There is a balance to be struck between inadequate referencing and the cluttering up of the text with an excess of repeated reference callouts. It is clear that we and the reviewer feel that balance to lie in slightly different places.

- *36) Conclusion: same comment as for the Introduction. This has to be significantly widened/rewritten.* We have expanded the conclusions section as noted earlier in this reply.

- *37) L256-257: "Whether they will become. . .be seen": Sentences like this one are so generic that contain basically no information content. Please avoid throughout the text (i.e. be more specific, refer to previous literature, extend, etc).* The reviewer is correct in a sense that the sentence carries no information. In another sense the reviewer is wrong: it appears to us to be a valid point that we cannot yet state that events such as the one described are becoming commoner, but that we should be able to make such a statement in the future, once we have more data. Moreover, it is important to make this statement to prevent readers assuming that events of the type described provide a clear signal of climate change.

- *38) Authors contributions: Who did the work (this is not mentioned in this section)? In addition, please check if only "suggesting many improvements to the text" is sufficient to grant a co-authorship in a scientific paper. This has to be checked with the ACP Editor as it is generally considered a very important ethical issue.* We have made the contributions section more explicit in a number of ways. In particular, we have stated explicitly that HCP did the bulk of the data analysis and modelling as well as writing the bulk of the text[1]. All of the authors have contributed in some way; we do not intend to remove any of them from the author list.

**Other changes**

- A reader (Albert Ansmann from Leipzig) has contacted us to note a couple of missing references. They were different missing references from those identified by reviewer 2. We have added these references to the final paper. One is Baars et al., ACP, (2019), doi: 10.5194/acp-19-15183-2019. The other is Torres et al. (2000), doi: 10.1029/2020JD032579.

- We note that part of the coastline of Alaska is missing from Fig. 3 in the paper. We have corrected this. We have also changed the symbol used for 46 hPa from an asterisk to a square. This is to ensure that it does not resemble the symbols for 147 and 100 hPa superimposed.

- In the light of the appearance of Lestrelin et al. (2021) (doi:10.5194/acp-21-7113-2021) we have added references to it and made additions to Fig. 4 for consistency with it in terms of nomenclature.

- We have added the GFS analysis data to the "Code and Data availability" section.
* * *
[1]We note that `latexdiff` does not mark up changes made to the author contribution section.

---

## Referee Report (RR1)

**Review of the manuscript ''Stratospheric pollution from Canadian forest fires" (now "MLS observations of biomass-burning products in the stratosphere from Canadian forest fires in August 2017") by Hugh C. Pumphrey et al., second stage.**

Dear Editor, Authors,

I have reviewed the resubmission of the manuscript ''Stratospheric pollution from Canadian forest fires" (now "MLS observations of biomass-burning products in the stratosphere from Canadian forest fires in August 2017") by Hugh C. Pumphrey et al., after the first stage of review and the subsequent Authors' modifications of the previous version. During my first review stage, I pointed at a number of issues, including two major comments. The main point was (and is) the fact that the paper looks, at least to my eyes, as very confusing and superficial in places and, I suppose, rushed. For what I see, even if some critical points have been solved (e.g. now the title is much less generic, the trajectory analyses are clearer, the "Authors contributions" section is more complete) many of my comments have been rejected by the Authors and/or unsuccessfully tackled. Thus, from my point of view, the two major issues have only been partially solved and so, unfortunately, I cannot recommend the manuscript for publication in its present shape. As I am a bit puzzled by the fact that the Authors decided to dismiss many of my comments, I leave it to the Editor to judge if a Major Revision or another decision is the best choice (I suggest Major Revisions). As a guide for solving the mentioned issues – in case the Editor thinks it useful -, please find some elements in the following. Please consider that this is not supposed to be a complete list of the modifications needed, this is just a set of examples of unclear/unprecise bits of text. Besides these points, I strongly suggest the Authors to thoroughly revise the text to improve clarity, preciseness and completeness.

Major Comments:
1) My comment during the first stage of the review process: '' The originality of the manuscript must be clarified and openly discussed, in particular with respect to the previous work of Yu et al., 2019.  What's new and what's complementary with respect to Yu et al., 2019? This is not clear at all in the present manuscript version, at least to me.''  The Authors decided that this is not necessary because "Given that our paper references Yu et al. (2019) and notes which species are described in that paper it did not seem necessary to us to go into more detail." I disagree on this point but might accept the small modification at L39-40, that implicitly mention the difference and complementarity of the two papers. It could (and should) be expanded a bit, so to be clearer: please consider this possibility.
2) I pointed at this, also: "The paper is unfortunately very confusing and certainly not very well crafted.  The Abstract, Introduction and Conclusions are very short, incomplete, probably rushed. Many statements are not justified but just "written down".  Many discussions and argumentations are just lacking.  All the analyses with trajectories (Sect.  4.2) are very obscure and must be extensively clarified/rewritten. Many crucial references are missing. The Authors should put a much larger effort in the writing of the text, the production of intelligible figures and the discussion of their results in the context of previous literature." I commend the change in the title, the efforts in extending the Abstract, the Introduction and Conclusions and the new organization with a very welcome "Results" unitary section. Nevertheless, this is not

yet satisfactory, in my opinion, and many of my specific comments have been either superficially dealt with or even just dismissed. I don't think that a Reviewer should need to insist on this aspect: writing a clear, accessible, unambiguous and complete manuscript is one of the main tasks of the Authors. By the way, a selection of most urgent specific comments, as examples of possible improvement of the text, is in the following (please revise the manuscript beyond these comments):

a. Abstract:
  i. "Events such as this are rare": "such as this" in which sense?
  ii. "third of four", "preceding two events…like the most recent": this is obscure at best. Please mention the specific events you're comparing to.
  iii. "Unlike the preceding two events (sic), but like the most recent event (sic), the polluted airmass described here had an unusually high water vapour content": how much? Why not putting specific results in an Abstract?
  iv. "these are in roughly the ratios to CO reported elsewhere": what does this means? Elsewhere where? Please use more words to express your ideas.
  v. "We use back-trajectories…": in Sect. 3.1, you have much more precise origin identification than just "originated in British Columbia fires" (also mentioned in the Conclusions): why a sentence so generic and superficial in the Abstract?

b. Introduction:
  i. "an important natural component": what do you mean with "component"? Please use a less generic word.
  ii. What do you mean with "nature" of fires? Again, too generic.
  iii. By the way, the full sentence is unclear to me. What do you mean with "even when…the fires"?
  iv. "more damaging": this is another generic statement I don't understand. You mean in terms of burned area? Of environmental/atmospheric impacts? Other?
  v. Please use reference in chronological order (e.g. Torres et al./Kloss et al.)
  vi. Kloss et al. is cited in a ACPD version and not the final ACP version. I did not check in the whole text but please double check if there are other occurrences of this.
  vii. "most detailed description" in terms of what?

c. Others:
  i. Specific comment 13 of first stage: "L34-37: Please justify these statements.", reply: "We regard the reference to the MLS data quality document to be sufficient for this purpose." It would be very easy to briefly mention the reasons why the use of lower levels are not recommended; a paper should be as self-sufficient as possible on these aspects, especially when referencing a technical document. Please expand.
  ii. Specific comment 14: "L40-41: "this is about. . .value": Where are these values (zonal mean; daily maximum) taken from?", reply

"From the MLS data at times outside of the event. We have made the wording of this sentence even more explicit." If I understand well, the "more explicit wording" is the following: "for this latitude band, for times immediately before the PNE". This is not explicit at all, in my opinion. Please specify this reference period, please.

iii. L130: "some results are shown in Khaykin et al., 2020": I don't see how a sentence of this type, "some results" can be part of a scientific paper. Which results? During the first review stage I pointed at, and I still do at the second stage, the superficial writing style in this manuscript, which is well represented by this example. I strongly suggest the Authors to look through the manuscript for the many points where this kind of lack of precision and details occurs. A detailed revision of this point is surely well beyond the scopes of a Referee.

iv. Specific comment 22: it is acceptable to cite open-source softwares these days but this is a scientific paper so, besides citing the nlm() function and R, please briefly explain how the errors are calculated.

---

## Author Response (AR2)

This version typeset on August 17, 2021
Note: We use *italics* for direct quotes from the reviewers.

**Reviewer 1**

Reviewer 1 required no further corrections.

**Reviewer 2**

Reviewer 2 gives the following introduction to his specific comments:

*I have reviewed the resubmission of the manuscript "Stratospheric pollution from Canadian forest fires" (now "MLS observations of biomass-burning products in the stratosphere from Canadian forest fires in August 2017") by Hugh C. Pumphrey et al., after the first stage of review and the subsequent Authors' modifications of the previous version. During my first review stage, I pointed at a number of issues, including two major comments. The main point was (and is) the fact that the paper looks, at least to my eyes, as very confusing and superficial in places and, I suppose, rushed. For what I see, even if some critical points have been solved (e.g. now the title is much less generic, the trajectory analyses are clearer, the "Authors contributions" section is more complete) many of my comments have been rejected by the Authors and/or unsuccessfully tackled. Thus, from my point of view, the two major issues have only been partially solved and so, unfortunately, I cannot recommend the manuscript for publication in its present shape. As I am a bit puzzled by the fact that the Authors decided to dismiss many of my comments, I leave it to the Editor to judge if a Major Revision or another decision is the best choice (I suggest Major Revisions). As a guide for solving the mentioned issues — in case the Editor thinks it useful — please find some elements in the following. Please consider that this is not supposed to be a complete list of the modifications needed, this is just a set of examples of unclear/unprecise bits of text. Besides these points, I strongly suggest the Authors to thoroughly revise the text to improve clarity, preciseness and completeness.*

We find it difficult to address the reviewer's desire to have us revise the entire text for clarity. The corresponding author has revised the text many times now, and the other authors have read it and provided extensive corrections which have been incorporated into the text. We have addressed the reviewer's specific comments as we note below, and have made a number of small corrections to the text. While we are fairly sure that these would not be regarded as sufficient by the reviewer, it is hard to know what he would consider to be sufficient.

- *1)My comment during the first stage of the review process: "The originality of the manuscript must be clarified and openly discussed, in particular with respect to the previous work of Yu et al., 2019. What's new and what's complementary with respect to Yu et al., 2019? This is not clear at all in the present manuscript version, at least to me." The Authors decided that this is not necessary because "Given that our paper references Yu et al. (2019) and notes which species are described in that paper it did not seem necessary to us to go into more detail" I disagree on this point but might accept the small modification at L39–40, that implicitly mention the difference and complementarity of the two papers. It could (and should) be expanded a bit, so to be clearer: please consider this possibility.* The paper by Yu et al. is about

almost entirely different aspects of the event than those covered by our paper. We have added a sentence which spells this out in more detail than before.

- *2)I pointed at this, also: "The paper is unfortunately very confusing and certainly not very well crafted. The Abstract, Introduction and Conclusions are very short, incomplete, probably rushed. Many statements are not justified but just "written down". Many discussions and argumentations are just lacking. All the analyses with trajectories (Sect. 4.2) are very obscure and must be extensively clarified/rewritten. Many crucial references are missing. The Authors should put a much larger effort in the writing of the text, the production of intelligible figures and the discussion of their results in the context of previous literature." I commend the change in the title, the efforts in extending the Abstract, the Introduction and Conclusions and the new organization with a very welcome 'Results' unitary section. Nevertheless, this is not yet satisfactory, in my opinion, and many of my specific comments have been either superficially dealt with or even just dismissed. I don't think that a Reviewer should need to insist on this aspect: writing a clear, accessible, unambiguous and complete manuscript is one of the main tasks of the Authors. By the way, a selection of most urgent specific comments, as examples of possible improvement of the text, is in the following (please revise the manuscript beyond these comments)* This is the kind of comment which is difficult to satisfy, and impossible to know whether you have done so. We have attended to the specific comments given below. But this general comment is so open-ended it is not possible for us to know whether we have improved the paper to the reviewer's satisfaction.

- *a. Abstract*

  - *i. "Events such as this are rare": "such as this" in which sense?* We felt that the wording used followed naturally from the previous sentence and was not particularly ambiguous. The reviewer is clearly an adherent of lucidity in scientific writing as set out by Michael McIntyre[1], and he is right to be so. We have attempted to re-word the offending sentence to remove ambiguous pronouns.

  - *ii. "third of four", "preceding two events...like the most recent" this is obscure at best. Please mention the specific events you're comparing to.* We have added a sentence noting that all of the other events occurred in Australia and have given the dates.

  - *iii. "Unlike the preceding two events (sic), but like the most recent event (sic), the polluted airmass described here had an unusually high water vapour content": how much? Why not putting specific results in an Abstract?* The main text gives a value of 14 ppmv where the background values are 3.5–5 ppmv. In other words, the plume values are 2.5–5 times the background value. We have altered the abstract to state this explicitly.

  - *iv. "these are in roughly the ratios to CO reported elsewhere": what does this means? Elsewhere where? Please use more words to express your ideas.* We have changed "elsewhere" to "in the literature". We are not sure what the reviewer wants here. The relevant references are in the main text and it is not usual to put references into an abstract.
* * *
[1]See `http://www.damtp.cam.ac.uk/user/mem/lucidity-in-brief/`

- v. *"We use back-trajectories...": in Sect. 3.1, you have much more precise origin identification than just "originated in British Columbia fires" (also mentioned in the Conclusions): why a sentence so generic and superficial in the Abstract?* We have added a specific location to the abstract and stated it as part of the results in the relevant section of the paper. A few numerical typos were corrected in the process.

- b. *Introduction:*

  - i. *"an important natural component": what do you mean with "component"? Please use a less generic word.* Sentence changed to use "process".

  - ii. *What do you mean with "nature" of fires? Again, too generic.* A typical example is that forestry practices can, by suppressing the natural fires, allow combustible material to build up on the forest floor. When a fire does occur, it then may be larger and more intense than the fires which would have occurred in a less managed forest. We have replaced "nature" with "intensity" — it is the only fix we could think of without adding a large digression.

  - iii. *By the way, the full sentence is unclear to me. What do you mean with "even when...the fires"?* We have had a go at making the sentence more specific.

  - iv. *"more damaging": this is another generic statement I don't understand. You mean in terms of burned area? Of environmental/atmospheric impacts? Other?* The sentence deliberately does not spell that out because it is speculating about the future rather than describing the past.

  - v. *Please use reference in chronological order (e.g. Torres et al./Kloss et al.)* Done, both here and wherever more than one paper are cited together.

  - vi. *Kloss et al. is cited in a ACPD version and not the final ACP version. I did not check in the whole text but please double check if there are other occurrences of this.* We have corrected this and apologise to the reviewer and his co-authors on that paper for the error. We have checked the BibTeX database used for "Discussion" papers and have, we think, updated any that are cited in the current paper.

  - vii. *"most detailed description" in terms of what?* In terms of "the evolution in time and space of the polluted airmass", as the remainder of the sentence spells out.

- c. *Others:*

  - i. *Specific comment 13 of first stage: "L34–37: Please justify these statements.", reply: "We regard the reference to the MLS data quality document to be sufficient for this purpose." It would be very easy to briefly mention the reasons why the use of lower levels are not recommended; a paper should be as self-sufficient as possible on these aspects, especially when referencing a technical document. Please expand.* We have attempted to give the reasons briefly without being led into a long digression from the topic of the paper.

  - ii. *Specific comment 14: "L40-41: "this is about ...value": Where are these values (zonal mean; daily maximum) taken from?", reply "From the MLS data*

*at times outside of the event. We have made the wording of this sentence even more explicit."* If I understand well, the "more explicit wording" is the following: "for this latitude band, for times immediately before the PNE". This is not explicit at all, in my opinion. Please specify this reference period, please. We have re-done the analysis, being more specific about the reference period and latitude band. The values change slightly, the essence does not.

– *iii. L130: "some results are shown in Khaykin et al., 2020": I don't see how a sentence of this type, "some results" can be part of a scientific paper. Which results? During the first review stage I pointed at, and I still do at the second stage, the superficial writing style in this manuscript, which is well represented by this example. I strongly suggest the Authors to look through the manuscript for the many points where this kind of lack of precision and details occurs. A detailed revision of this point is surely well beyond the scopes of a Referee.* Because Khaykin et al. (2020) is about a different event from the current paper, we did not want to distract the reader by going into any detail of their results. However, we felt that it was necessary to mention those results briefly to show that there is no fundamental reason why the method we describe for CO could not be used for other species, if the amount of that species injected by an event was large enough. It was clear to us that the second part of the sentence under discussion follows on from the first part. We have added three words to make this more explicit, but we do not think it necessary to expand the sentence any further.

With regard to the referee's broader point we note that the corresponding author has already had the paper read and corrected in detail by the co-authors.

– *Specific comment 22: it is acceptable to cite open-source softwares these days but this is a scientific paper so, besides citing the nlm() function and R, please briefly explain how the errors are calculated.* We have added a couple of sentences in an attempt to explain how the errors are calculated.

**Other changes**

A reference was added to a paper by one of the co-authors (M. D. Fromm) which was published while the second round of corrections to this paper were ongoing. A sentence was added to note that although the first obviously-enhanced CO values were observed on 14 August, Fromm et al (2021) report values at the high end of the normal range on 13 August at a location where CALIOP shows aerosol from the PNE. A further sentence was added in order to reference a paper by Das et al., which was also published while the current paper was being revised.

---

## Author Response (AR3)

This version typeset on September 9, 2021

Note: We use *italics* to quote the editor's comments.

- *Title: I would suggest to write "Microwave Limb Sounder (MLS)" instead of just "MLS"* Done

- *General comment on the manuscript. On several occasion you say the numbers are arbitrarily chosen. This needs more motivation why the given numbers have been chosen. I guess these are numbers you expect to be correct or suitable for your analyses. So that should be clearly stated and it also should be mentioned how did you derive these or how you can assume that this values could be representative.* A co-author points out that the choices we have made are not really arbitrary in the dictionary sense : "capricious, random, on whim, . . .". Where the word appears, I have replaced it with a more suitable one. Searching through the text, I find three places where a choice was made which we previously described as arbitrary:

  - Reasons are given for the choices for the value of $\alpha$ in sec 3.1.1, so I have left this unchanged, other than removing the words "somewhat arbitrary".

  - The decision to use a constant, plus annual and semiannual cycles to represent the CO background is described as "arbitrary"; I have changed this to "informed. We already state that we consider that other choices would give similar results.

  - P16, L264 – 265: We described the choice of the fixed size of the release box as "somewhat arbitrary"; we have removed these words. We already explain that the chosen size reflects the size of the observed pyroCb clouds.

  In all these cases, we already give reasons for the choice.

- *P1, L9: that → than* "That" is actually what I meant, and "than" on its own would not make sense. I have gone for . . . 5 times greater than that in . . .

- *P1, L11: "these are in roughly the ratios to CO........" this sentence makes no sense, please rephrase/correct.* Sorry about that. It seemed to make sense to me and to those co-authors who read the draft in any detail. This means that it is hard for me to know what wording would be clearer. The longer explanation is that

  - Measurements have been made by various other workers of the mixing ratios of CO and various other gases in various other biomass-burning plumes.

  - We have calculated the ratio of (say) HCN mixing ratio to CO mixing ratio in those other plumes.

  - The values of [HCN]/[CO] from the MLS data for the PNE are similar to the values of [HCN]/[CO] from those other plumes — here I am using [] to indicate volume mixing ratio.

  I thought I had encapsulated that in a single clear sentence suitable for an abstract, but clearly I had not succeeded as well as I thought I had. I have gone for an alternative which is, I think, easier to follow, but less explicit.

- *P1, L19: Full stop before reference of Williams and Abatzoglou obsolete.* Fixed.

- *P3, L28 [68 actually]: Parentheses around Fromm et al. obsolete (should be around the given year).* There were two sets of parentheses: one around the entire sentence and one around the year in the reference. As per your email reply to my question about this, I have removed the parentheses round the whole sentence.

- *P3, L69: closing parentheses obsolete.* This is the close parenthesis at the end of the sentence above, so I have removed it.

- *P8, L128: table 1 → Table 1* Fixed.

- *P12, Table 3 caption: error → either use plural, thus "errors" or write "the error".* I chose "errors".

- *P13, L197: The relationships with CO of CH3CN.........? What do you exactly mean? The relationships "of" CO "to/with" CH3CN? Please check and correct.* I have made the sentence longer, but hopefully more explicit.

- *P13, L227ff: I don't like the term "launched" for trajectories. I would prefer "started".* "launch" replaced with "start" in all trajectory contexts.

- *P15, Figure 11 caption: at altitude 12 km -¿ either write "at an altitude of 12 km" or just "12 km"* Fixed.

- *P16, L264 – 265: remove parentheses around the sentence.* Done.

- *P17, L293: Add a reference?* This is the line that says " Most forest fires do not produce a pyroCb cloud. Moreover, most pyroCb clouds do not extend to a great enough altitude to loft the products high enough for MLS to observe them." I could not find a reference that spells the content of this sentence out. I have therefore added an extra few sentences referencing Peterson et al. (2017) for a typical number of pyroCbs per year (much larger than 1/17) and two sources for typical numbers of fires per year (thousands). Much of this was at the suggestion of my co-author Mike Fromm.

- *P18, L315: There is only a year given, but no author name or other source.* True. We are supposed to reference datasets these days, preferably using the DOI. I have done that, and I have used the BibTeX entry provided on the web pages to which the DOI points. That BibTeX entry simply gives the author as `author={NASA/LARC/SD/ASDC}` — it does not provide any human names.

- *P18, L320: the areas burned → the areas that burned?* Fixed here and in the relevant figure caption.

- *P19, 343: Change brackets to parentheses and move the references after "Simulations" at the end of the sentence.* Fixed.

- *P20, L363: Better to write "number" instead of "figure" (even if then appears twice in the sentence)?* Fixed.

- *P20; last sentence of the conclusion: What about the IPCC report or other climate change studies? If the extreme weather conditions will occur more frequently one also could expect forest fires to occur more frequently.* We consider that prediction

to be hard to justify because PNE-like events require conditions other than the presence of a fire. Rather than expand the conclusions (which really should not contain new material not covered earlier in the paper) I have added a few sentences to the discussion, together with a reference to Williams and Abatzoglou (2016). The last sentence of the conclusion is unchanged, but now exists to summarise the new sentences in the discussion, along with their reference. (I avoided referencing the new IPCC report as many sections of it still say "DO NOT QUOTE OR REFERENCE".)

In other changes I have added the editor and the two reviewers to the acknowledgements. I intend to add "The article processing charges for this open-access publication were paid by the RCUK Open Access Publication Fund." This assumes that they agree to pay the charges.